# Emergent dynamics due to chemo-hydrodynamic self-interactions in active polymers

Manoj Kumar [1] ✉, Aniruddh Murali [1], Arvin Gopal Subramaniam[2], Rajesh Singh [2] ✉ & Shashi Thutupalli [1,3] ✉

The field of synthetic active matter has, thus far, been led by efforts to create point-like, isolated (yet interacting) self-propelled objects (e.g. colloids, droplets, microrobots) and understanding their collective dynamics. The design of flexible, freely jointed active assemblies from autonomously powered sub-components remains a challenge. Here, we report freely-jointed active polymers created using self-propelled droplets as monomeric units. Our experiments reveal that the self-shaping chemo-hydrodynamic interactions between the monomeric droplets give rise to an emergent rigidity (the acquisition of a stereotypical asymmetric C-shape) and associated ballistic propulsion of the active polymers. The rigidity and propulsion of the chains vary systematically with their lengths. Using simulations of a minimal model, we establish that the emergent polymer dynamics are a generic consequence of quasi two-dimensional confinement and auto-repulsive trail-mediated chemical interactions between the freely jointed active droplets. Finally, we tune the interplay between the chemical and hydrodynamic fields to experimentally demonstrate oscillatory dynamics of the rigid polymer propulsion. Altogether, our work highlights the possible first steps towards synthetic self-morphic active matter.

A fundamental challenge in the study of active matter is the detailed understanding and tuning of the interplay between mechanics and chemical (phoretic) fields. The effect of self-generated chemical fields on the dynamics of the constituent units is a subject of intense study[1–5]. Such an interplay arises both in the context of living systems[6–8] and synthetic non-living emulations[9–14]. Experimentally, the emergent consequences of chemo-mechanical self-coupling in synthetic active matter systems comprised of point-like unit active particles have been reported[12,13,15]. However, the inspiration for active systems, biology, is replete, not only with examples of point-like active units, but also higher order (hierarchical) assemblies of active units, that form extended objects. These assemblies display remarkable emergent dynamics and self-organization, be it the mechanical consequences of

active polymers such as actin and microtubules, the folding and function of coded polymers such as proteins or organismal shape arising from embryonic tissues. These biological instances may be viewed as self-morphing matter—assemblies of (chemically) active subunits—the dynamical traits of which are an intricate choreography between mechanics and other self-generated guiding fields, such as chemistry.

The study of assemblies of active synthetic units—such as molecules[16], polymers[17–20] and sheets[21], is only a recently growing area of research. Some experimental realisations of such systems have been achieved by using *external sources of driving* such as electro-magnetic actuation[22–27]. This kind of external driving mechanism stands in contrast to active and living systems, in which the internal chemistry drives the system out-of-equilibrium. This difference between external and

[1]Simons Centre for the Study of Living Machines, National Centre for Biological Sciences, Tata Institute of Fundamental Research, Bangalore, India. [2]Department of Physics, Indian Institute of Technology, Chennai, India. [3]International Centre for Theoretical Sciences, Tata Institute of Fundamental Research, Bangalore, India. ✉e-mail: manojk@ncbs.res.in; rsingh@physics.iitm.ac.in; shashi@ncbs.res.in

internal driving is crucial—since the control fields are imposed externally, the interactions between the monomers are not self-responsive, leading to a fundamental difference in the emergent dynamics[1,2,8]. Although chemically self-interacting catalyst-coated colloids that are catalytically actuated locally have been used to construct linear polymers[28,29], the "monomeric" units are neither self-propelled nor are they orientationally free. Emulsion droplets, due to their internal fluidity offer a promising monomeric unit of study. Previous work on constructing linear assemblies and freely-jointed chains using emulsion droplets have used sticky DNA linkers[30–33], though these chains were passive. However, active-droplet-polymers offer a well-controlled, simplified, microscopic system, wherein each constituent monomer unit exhibits autonomous motion within the fluid medium. The soft interface of these droplets (compared to the solid interface of colloids) facilitates facile chemical modification of their interfaces thereby enabling unfettered rearrangement of the droplets within a chain. Despite binding, the free rearrangement of the droplets in an assembly can be exploited even to create a flexible, foldable structures[34]. Nevertheless, the design of chemically-linked, flexible, and freely jointed active polymers from autonomously powered components remains a challenge.

In this study, we construct freely jointed chemically-interacting active colloidal polymers, and study their dynamics. The propulsion mechanism of the monomers (emulsion droplets) within the chain creates external chemical and hydrodynamic fields, causing self-shaping interactions within the polymer. We measure these chemical and hydrodynamic fields and quantify, as a function of the polymer lengths, the emergent rigidity, shapes and ballistic self-propulsion of the active polymers in quasi two-dimensional confinement. We show that a model which includes only chemical interactions between the monomers captures all these activity-driven conformational features quantitatively, thereby establishing that the minimal requirements for the emergent self-organization are: (i) the monomer droplets of the

chain propel due to (gradients of) their self-generated chemical field; and (ii) the structures are confined to move in quasi two-dimensions. Finally, by tuning the coupling between the chemical and hydrodynamic fields, we demonstrate oscillatory gaits of the polymers. Altogether, we envision these as initial steps towards a kind of synthetic active matter capable of emergent self-morphic dynamics.

## Results
### Freely jointed active polymers of self-propelling emulsion droplets

The self-propelled droplets that we use as monomers are comprised of oil, slowly dissolving into an external supramicellar aqueous solution of ionic surfactants[35,36]. The dissolution of the droplets results in the spontaneous development of self-sustaining gradients of surfactant coverage around the droplets. These gradients give rise to Marangoni stresses causing the droplets to propel[37] (Fig. 1A). As such, these self-propelled droplets may be viewed as a physico-chemical realization of the so-called squirmer model for microswimmers, a sphere with a prescribed surface slip velocity that exchanges momentum with the fluid in which it is immersed[38–40]. The qualitative and quantitative nature of this slip velocity governs the near and far-field hydrodynamic flow perturbation around the squirmer and hence its self-propulsion. The hydrodynamic flow fields around the oil droplet microswimmer (Fig. 1B, Supplementary Video SV1) depend in tunable ways on geometry also, as we show later, chemical conditions[36,41–43]. In addition to the hydrodynamic flows, these microswimmers leave a trail of chemical fields[12]—comprised of oil-filled surfactant micelles formed by the transfer of oil molecules into empty surfactant micelles—which can be visualised using an oil-soluble fluorescent dye (Fig. 1C, Supplementary Video SV1). The trail persists because the oil-filled micelles take longer to diffuse than the surrounding oil-free surfactant micelles and hence cause a remodeling of the environment around the droplets, leading to time delayed negative chemotatic self-interactions[12].

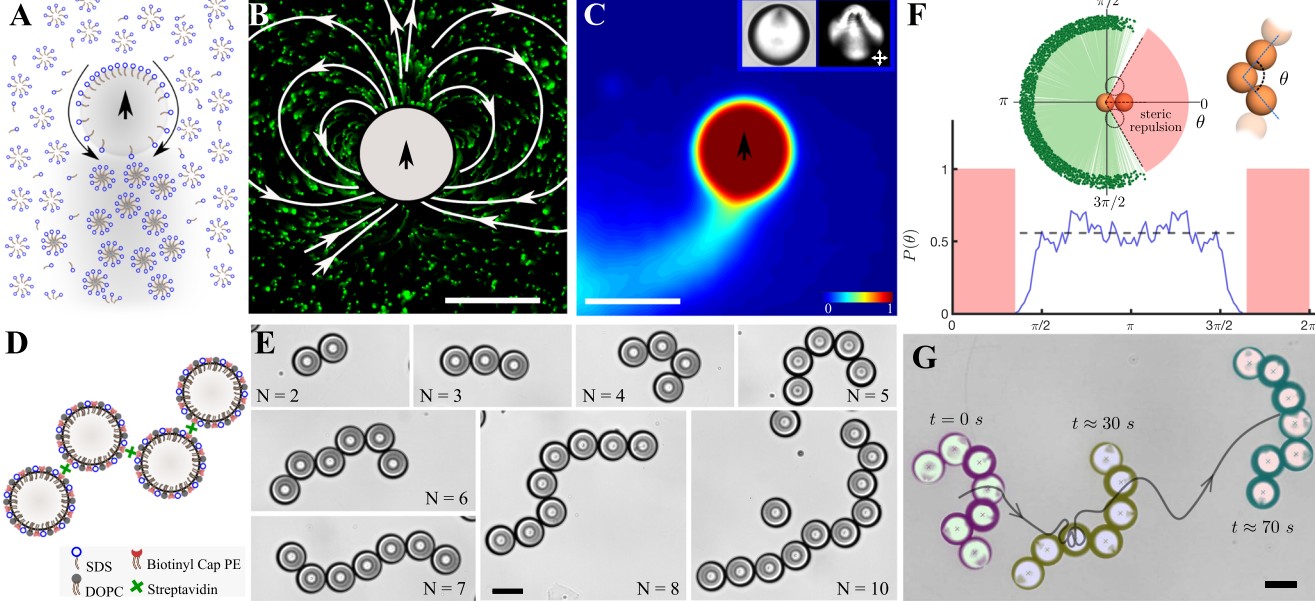

**Fig. 1 | Self-propelled droplets are used as monomers to assemble freely-jointed active polymers. A** A monomeric liquid crystal droplet in a surfactant micellar solution, propels due to a spontaneously generated surfactant gradient at its interface. **B** The propulsion of the nematic liquid crystal 5CB droplets results in characteristic hydrodynamic flow fields and **C** chemical fields visualised using oil-soluble fluorescent (Nile red) dye mixed with 5CB. *Inset*: the symmetry breaking associated with the self-propulsion and the defect structure within the nematic droplet is apparent in the bright field and cross-polarised images of the self-propelling droplets. **D** Scheme for assembling linear polymers of the active droplets using biotin–streptavidin chemistry. **E** Linear, chemically linked, inactive N-meric chains of 5CB emulsion droplets, comprised of increasing numbers of monomer units. **F** The bond angles of the polymer exhibit a uniform distribution i.e. the polymers are freely-jointed (number of chains n = 9, 2200 data points). **G** When activated, i.e. immersed into surfactant solutions of high enough micellar concentration, the polymers exhibit active motion. The image shows three time points of the polymer, overlaid with the center of mass trajectory, moving in a weakly confined setting. All scale bars represent 50 μm. Source data are provided as a Source Data file.

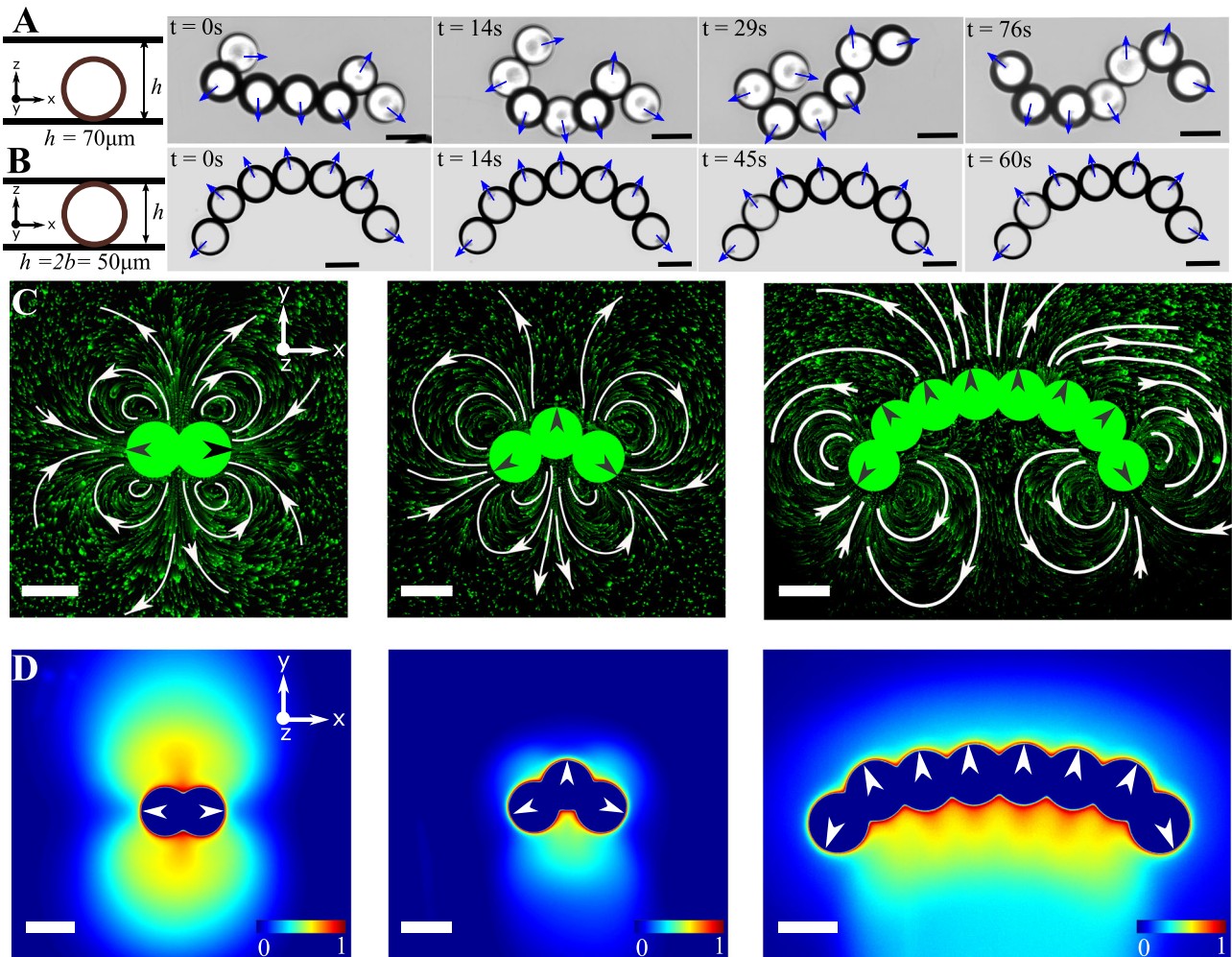

**Fig. 2 | Chemo-hydrodynamics of the active polymers in strong confinement.** Time-dependent configurations (*left* to *right*) due to the activity of the chains in **A** weak confinement ($h/2b \approx 1.4$) and in **B** strong confinement ($h/2b \approx 1$), where $b$ represents the droplet radius and $h$ denotes the confinement height ($h \approx 50$ μm) (scale bar: 50 μm). Experimentally measured **C** hydrodynamic flow fields and **D** chemical fields for the situations corresponding to the strong confinements as in **B**. From *left to right*: dimer, trimer, and octamer (scale bar: 50 μm, colors of the chemical fields indicate normalized values of the concentration, with blue being the lowest and red being the highest).

To create tethered assemblages, we use biotin-streptavidin chemistry to form freely jointed chains of the active droplets (Fig. 1D, Materials and Methods). We use a hybrid (surfactant-lipid) monolayer of surfactant and biotinylated lipids that coat the droplets spontaneously (Supplementary Information, SI Fig. S1A, B). Using these biotinylated lipids in conjunction with the well known specific interaction of biotin with streptavidin, we "polymerise" the droplets to form linear chains via controlled assembly and incubation (details in Materials and Methods, and Supplementary Information, SI Fig. S1B). This assembly process robustly results in linear assemblies of 5CB oil emulsion droplets such as dimers, trimers, and so on up to decamers (Fig. 1E). Our protocol rarely yields chains longer than $N = 10$ without branching, and the longest linear chains we obtain consist of $N = 13$ droplets (Fig. 1E and SI Fig. S1D). Given that the assembly process is a one-pot reaction in equilibrium, the monomers link with each other only probabilistically in time; intuitively, therefore the chance of branching of linear chains increases with increasing chain length. Notably, we also observed branching even for shorter assemblies (monomers $N \leq 10$), but the yields of linear chains are higher and they are also easily separable from the branched chains and free monomers in our protocol. The resultant polymers are freely-jointed, as seen from the distribution of the internal bond angles between the monomer droplets (Fig. 1F), only restricted by the steric interaction between the droplets. Furthermore, since these are liquid droplets, there are also convective flows that are generated inside these self-propelled oil droplets[44] which can cause dynamic reorientation of the propulsion direction (this is in contrast to asymmetric colloids such as Janus colloids, whose orientation with respect to the polymer axis remains fixed by design). We activate the chains by immersing them in 25 wt% SDS surfactant solution. This causes the monomers within the chain to propel (Fig. 1G) and thereby the entire polymer is set into motion. Generically, this propulsion is three-dimensional, and for ease of visualisation and analysis we only focused on confined settings (using Hele-Shaw like geometries) in this paper (SI Fig. S1E). The fully flexible active polymer dynamics are evident in the partially three-dimensional random motion and orientation of the polymer chains (Fig. 1G and Supplementary Video SV2).

## Self-shaping chemo-hydrodynamic interactions within the active polymer in quasi two-dimensional confinement

When the active polymers are confined to a quasi two-dimensional setting—a Hele-Shaw cell—the chains no longer exhibit flexible motion but rather become *rigid* and adopt a stereotypic C-shape (Fig. 2A vs. B, Supplementary Video SV3). To gain an understanding of the polymer dynamics, specifically in strong confinement,

*i.e. $h/2b \approx 1$*, we measured the hydrodynamic and chemical fields produced by active polymers of different lengths in these conditions. Some of these fields are shown in Fig. 2C and D (see also Supplementary Videos SV4–SV9). The flow fields are measured using standard microPIV techniques with fluorescent tracer particles of diameter $0.5\,\mu m$ pre-mixed in the surrounding continuous medium. The chemical fields on the other hand are visualized using an oil-soluble fluorescent dye (Supplementary Experimental Section). When the oil droplets dissolve via micellar solubilization, micelles filled with dye-doped oil are formed, enabling the visualisation of the field of filled-oil-micelles (the chemical field) around the monomer droplets in the chain (Fig. 2D). The hydrodynamic flow fields around the chains are not merely a superposition of the field around a monomer (*Cf.* Fig. 1B), underscoring the non-linearity of the chemo-hydrodynamic coupling between the monomers within the chains (see Supplementary Information Fig. S4). The flow fields of polymer lengths of $N = 2$, 3 and 8 are shown here only as examples to point out generic features of these dynamics. There is a qualitative difference in the fields of a dimer from that of the other N-mers: the dimer attains a metastable symmetric configuration while the steady-state hydrodynamic and chemical fields of all other N-mers are asymmetric (see Supplementary Information Fig. S6A–H, Supplementary Videos SV4–SV9). The asymmetries in the chemical and hydrodynamic fields of the longer polymers reflect the stereotypic C-shapes and therefore a force imbalance that results in a net propulsion of the active polymers.

Next we seek to capture the quantitative features of the active polymer dynamics in strong confinement. A priori it may seem that such an understanding requires a model for this system that includes a solution of the chemical and hydrodynamic fields along with satisfying appropriate boundary conditions on the surface of the particles and at all other fluid-solid boundaries. However, such a full numerical solution satisfying those boundary conditions is a challenging task; instead, we seek a *minimal* model that captures the salient features from the experiments. Specifically, we focus on the emergence and nature of the stereotypic C-shape of the polymers and their self-propulsion dynamics, without any emphasis on recapitulating features of the flow and chemical fields.

## A minimal model for the active polymer dynamics in confinement

In building a minimal model, we begin by noting that in the far-field description, specifically for the kind of chemo-hydrodynamic swimmers as we have above, chemical fields decay slower than hydrodynamic fields in two dimensions[45]. On the other hand, to obtain the chemical and hydrodynamic fields near the droplets accurately, we need to resolve the near-field phoretic and hydrodynamic interactions[46]. Coupling the hydrodynamics with the chemical field along with appropriate boundary conditions would necessitate a full numerical solution, which we do not pursue here. Instead, we focus on a simple model of chemical interactions which accounts for trail mediated phoretic interactions. Indeed, such a rationalization has been applied recently in successfully describing the dynamics of active droplets[12]. Therefore in our model for the active polymers, we ignore hydrodynamic effects and account only for the chemical interactions. The chemical field around a monomer droplet is modeled by considering the *i*th monomer as a point source of the chemical field centered at $\mathbf{R}_i$, which self-propels with speed $v_s$ along the direction $\mathbf{e}_i$. The velocity and direction of the monomer can change due to chemical interactions. The position $\mathbf{R}_i$ and orientation, given by the unit vector $\mathbf{e}_i$, of the *i*th monomer is updated using the following kinematic equations:

$$\frac{d\mathbf{R}_i}{dt} = \mathbf{V}_i, \qquad \frac{d\mathbf{e}_i}{dt} = \mathbf{\Omega}_i \times \mathbf{e}_i. \qquad (1)$$

Here, the translational velocity $\mathbf{V}_i$ and angular velocity $\mathbf{\Omega}_i$ of the *i*th monomer are given as:

$$\mathbf{V}_i = v_s \mathbf{e}_i + \chi_t\,\mathcal{J}_i + \mu\mathbf{F}_i, \qquad \mathbf{\Omega}_i = \chi_r(\mathbf{e}_i \times \mathcal{J}_i). \qquad (2)$$

In the above equation, $\mu$ is the mobility of a monomer, while the chemical interactions between the monomers are contained in the vector $\mathcal{J}_i = -(\nabla c)_{\mathbf{r}=\mathbf{R}_i}$, where $c$ is the concentration of filled micelles. The concentration field of the filled micelles is obtained by considering each emulsion droplet as a point source of the chemical field (explicit expressions of $c$ and $\mathcal{J}_i$ are given in the Supplementary Information). The resultant interactions are thus said to be *trail-mediated*, where the chemical interactions at time $t$ depend on the historical trajectories (chemical trails) of all monomers in the chain at times $t' < t$ (see SI Eq. S2 for the explicit form). Implications of these will be explored further in the section "Metastable configurations exhibiting spontaneous polymer rotations". The constants $\chi_t$ and $\chi_r$ take positive values. $\chi_t > 0$ implies a repulsive chemical interaction between the monomers while they are held together by the attractive spring potential described below. The constant $\chi_r > 0$ implies that the monomers rotate away from each other in the freely joined chain. Such a reorientation is consistent with the negative auto-chemotactic behavior of the monomer droplets[41], unlike the chemical interactions of purely phoretic colloids. In the experiments, the droplets are freely joined using biotin-streptavidin chemistry, which is captured in the model by an attractive spring potential. The resulting spring force on the *i*th monomer is given as: $\mathbf{F}_i = -(\nabla U)_{\mathbf{r}=\mathbf{R}_i}$, where $U = \sum_{i=1}^{N-1} U^C(\mathbf{R}_i, \mathbf{R}_{i+1})$ and $U^C = k(r_{ij} - r_0)^2$ is spring potential of stiffness $k$ and natural length $r_0$ which holds the chain together. Here, $r_{ij} = |\mathbf{R}_i - \mathbf{R}_j|$ and the spring's natural length is equal to the diameter of the monomer used in the experiment, $r_0 = 2b$. In our experiment, the typical active force $[\mathcal{O}(6\pi\eta b v_s) \sim 10^{-11}N]$, with $\eta$ being the viscosity of the solvent is dominant to the typical Brownian forces $[\mathcal{O}(k_B T/b) \sim 10^{-16}N]$, and is thus ignored in our model. We simulate the model by integrating the above equations numerically. Simulation details are given in Supplementary Information (SI) along with the full list of parameters which is present in the Supplementary Information Table 1. The values of parameters $b$, $\eta$, $r_0$, and $v_s$ in simulations exactly match those from experiment. The susceptibilities $\chi_t$ and $\chi_r$ were chosen to best match the time scales of transitions from a transient state to a steady-state. The diffusivity $D_c$ is chosen to be about 50 times higher than the experimental value. Choosing a $D_c$ value identical to those from experiment also reproduces the phenomenology here, though the spatial curvature along the chain is much less pronounced—thus our choice effectively accounts for the additional effect of micellar advection. This is consistent with the fact that the advection of the micelles by the fluid flow is an important part of the physics of the problem. The role of micellar advection by the flow becomes more apparent in the results presented in the section "Time varying chemo-hydrodynamic fields and the emergence of periodic polymer oscillations". A detailed study of the role of these parameters and resulting steady-states for the system considered here, as well as for chemo-attractive and non-reciprocal chemical interactions, is explored in a follow-up work[47].

## Chemical interactions alone can account for the emergent rigidity and dynamics of the active polymer

We now compare the results obtained from the chemical model with the experimental findings. First, we characterize the C-shape of the polymers in strong confinement and show that shape is associated with the "stiffening" of the active polymer. To do so, we compute the "positional" and "orientational-rigidities" of the chains. The positional rigidity is quantified by the distribution of the positions of the monomers (in the center of mass frame) of the chain along the direction perpendicular to the direction of motion (Fig. 3A, inset of B and SI Fig. S1F). The

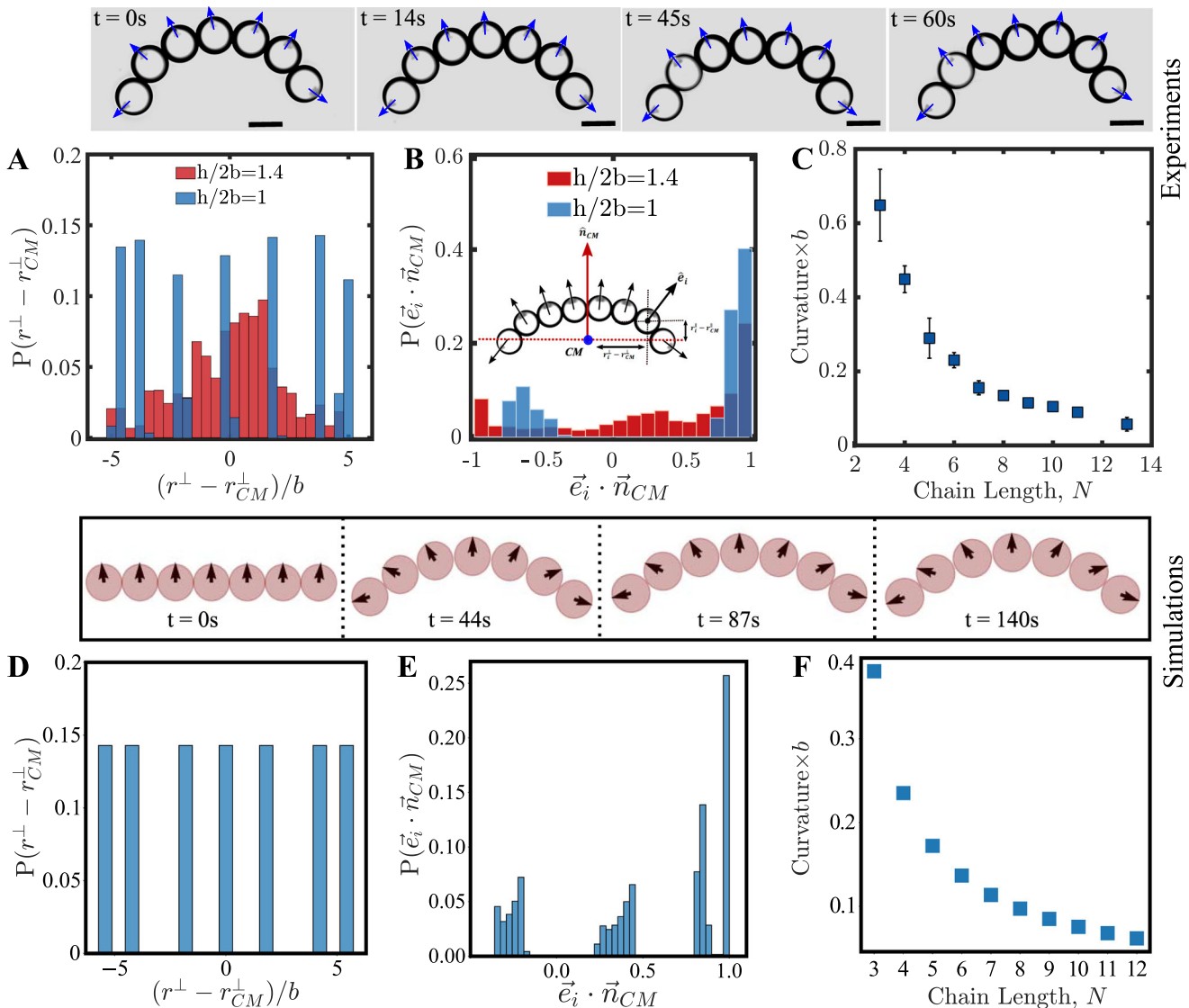

**Fig. 3 | Emergent rigidity of the polymer in strong quasi two-dimensional confinement.** Top panels correspond to experiments and bottom panels correspond to simulations. Experimental image snapshots are reproduced from Fig. 2 for clarity. **A**, **B** are the distributions of the positional and orientational degrees of freedom of the active polymer chain experimentally measured in weak confinement ($h/2b \approx 1.4$) and in strong confinement ($h/2b \approx 1$) for $N = 7$ (number of chain $n = 1$, data points $\geq 210$). The positional rigidity measured from the chain centre of the mass (*CM*) only in a direction perpendicular to the direction of propulsion; this is denoted $r^{\perp} - r_{CM}^{\perp}$. The orientational rigidity of the chain is measured as $e_i \cdot n_{CM}$ where $e_i$ is a unit vector which specifies the orientation of the *i*th monomer and $n_{CM}$ is the unit vector of the center of mass (*inset*, **B**). **C** Dimensionless mean curvature

(obtained by multiplying curvature with the droplet radius) of different polymer chain lengths is plotted against chain length (*N*) (see SI Fig. S7) (error bars represent ± SD). *Lower panel*: simulations results of the active polymer dynamics showing time snapshots of active chains, for chain sizes of $N = 7$. Black arrows indicate the orientation $e_i$ of the particles. The time-snapshots are for the chain steady states attained immediately from any arbitrary initial condition. **D**, **E** show the positional and orientational rigidities of the active chain ($N = 7$) obtained from simulations. **F** The dimensionless curvature of the active polymer in simulations as a function of the *N*. Source data for experiments are provided in the Source Data file and simulation code is provided via a Code Ocean link below.

orientational rigidity is obtained by computing the correlation between the orientations of the monomers with respect a direction orthogonal to the long axis of the polymer (Fig. 3B). In weak confinements ($h/2b \approx 1.4$), the histograms for the positional (Fig. 3A) and orientational (Fig. 3B) rigidities exhibit a broad distribution for the droplet positions and orientations indicating a freely-jointed or flexible nature of the chain. However, in strong confinement ($h/2b \approx 1$), we see distinct peaks for the positions and orientations of the monomers within the chains, indicating that these degrees of freedom are fixed in the system (quantified by the C-shape), and hence an effective rigidity of the polymer has emerged in strong confinement. Another measure of the positional rigidity of the chain is the curvature of the steady state chain, which we measure via

the Monge representation[48]. This displays an increase with the polymer length, suggesting greater stiffening of longer polymers (Fig. 3C).

Snapshots from simulations of the above model are shown in the lower panel of Fig. 3, where the polymer structures are confined to move in two dimensions (see also Fig. S7 in SI). Choosing an initial condition of a straight rod with parallel orientations, the rigid C-shape is relatively quickly obtained (within simulation time scales), after which the chain maintains its ballistic propulsion (see also Supplementary Video SV10). A more in-depth discussion on the effect of different initial conditions and the universality of the C-shape will be reported later in the section "Metastable configurations exhibiting spontaneous polymer rotations". This emergent C-shape in the steady

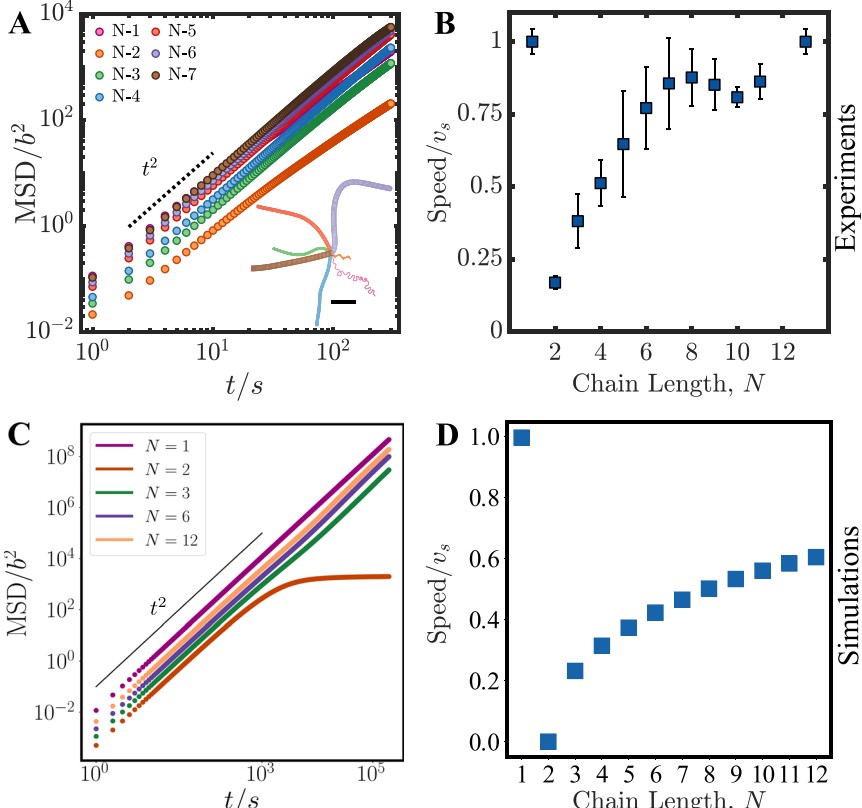

**Fig. 4 | Dynamics of the active polymers in strong quasi two-dimensional confinement. A** Mean squared displacement (MSD) profiles of experimental trajectories for different length chains ($N=1$ to $N=7$), represented in dimensionless form by dividing the MSD with $b^2$ and time with $1s$ where $b$ is the radius of the droplet (Supplementary Video SV4). Dashed line indicates the ballistic (MSD ~ $t^2$) limit. Inset represents the trajectories of different polymer chain lengths are shown in different colors (Supplementary Experimental Section, SI Figs. S2 and S3) (scale bar: 1 *mm*) (number of chains, n ≥10 averaged per chain length). **B** Normalised speed of different

polymer chain lengths ($N$) which increases with the chain length. The speed of a polymer chain is normalised with respect to the monomer speed, $v_s$ (Supplementary Experimental Section, SI Fig. S3)(error bars represent ± SD). Bottom panels **C** and **D** show corresponding results from simulations. C shows the MSD for different chain lengths, along with the ballistic limit of MSD ~ $t^2$ as reference. D Simulated speed of polymers in steady state, normalised by the speed of a single particle $v_s$, as a function of the chain length $N$. Source data for experiments are provided in the Source Data file and simulation code is provided via a Code Ocean link below.

state as seen in the experiment (Fig. 3 top panel snapshots and Supplementary Video SV3) is therefore reproduced in the simulations of the model. We also quantify the positional and orientational rigidities in the aforementioned manner. Strikingly, the results from our minimal chemical model show very similar features (steady-state shape, positional and orientational distributions–Fig. 3D, E). Further, the dependence of the polymer curvature with the length of chain, *i.e.* greater stiffening with increasing chain length is also found to match experimental observations (Fig. 3F; see Supplementary Information for details on computation of the curvature).

As also pointed earlier, this emergent rigidity associated with the fore-aft asymmetric C-shape of the active chains, is concomitant with ballistic propulsion of the polymers in a direction orthogonal to the length of the polymer. The propulsion dynamics are captured in the experiments by measurements of the mean-squared displacement (MSD) (Fig. 4A) and saturation speeds of the centers of mass of the chains, which varies with the number of monomeric units comprising the polymer (Fig. 4B and Supplementary Video SV11), with longer polymers moving faster than shorter ones. We note that while the speeds of the polymers are less than that of the isolated monomeric units, they asymptote to the instantaneous speed of a monomeric droplet with increasing polymer length (Fig. 4B). Notably, these dynamical experimental features of the active polymers are also captured by the model that we have developed. Any differences that exist are in the exact quantitative details (exact orientational distribution at the steady state–see Fig. 3A/B vs. D/E, as well as comparison of the

y-axis of Fig. 4A vs. C and Fig. 4B vs D) of our computed quantities. In addition, the aforementioned halting for the $N=2$ (dimer) is captured in the simulations, but this halting is not metastable as in the experiment.

Altogether, the agreement between the simulations and the experiments affirms, *post facto*, the choice of our minimal model and indicate that the polymer rigidity and dynamics emerge from the following ingredients: (i) trail-mediated chemical interactions, and (ii) two-dimensional confinement, with hydrodynamics potentially giving rise only to higher order quantitative corrections.

## Metastable configurations exhibiting spontaneous polymer rotations

To emphasise the suitability of our minimal modeling approach, we now show that the chemical interactions can also capture the dynamics of other metastable shapes of the active polymers. As discussed earlier, the chains adopt a stable C-shape self-propelling configuration. In experiments, we can further observe transitions from other shapes to the C-shape. To observe such shapes, we note that the C-shape can be destabilised in two ways: (i) by increasing the height of confinement such that the droplets forming the chain can sample the full three dimensional space (e.g. Figs. 1F, 2A and Supplementary Video SV2); and (ii) by direct-collision or interactions with surrounding assemblies (e.g. Supplementary Video SV12). In the experiments, we often see such destabilisation via the latter mechanism, and this causes the chains to transition from the C-configurations to a metastable

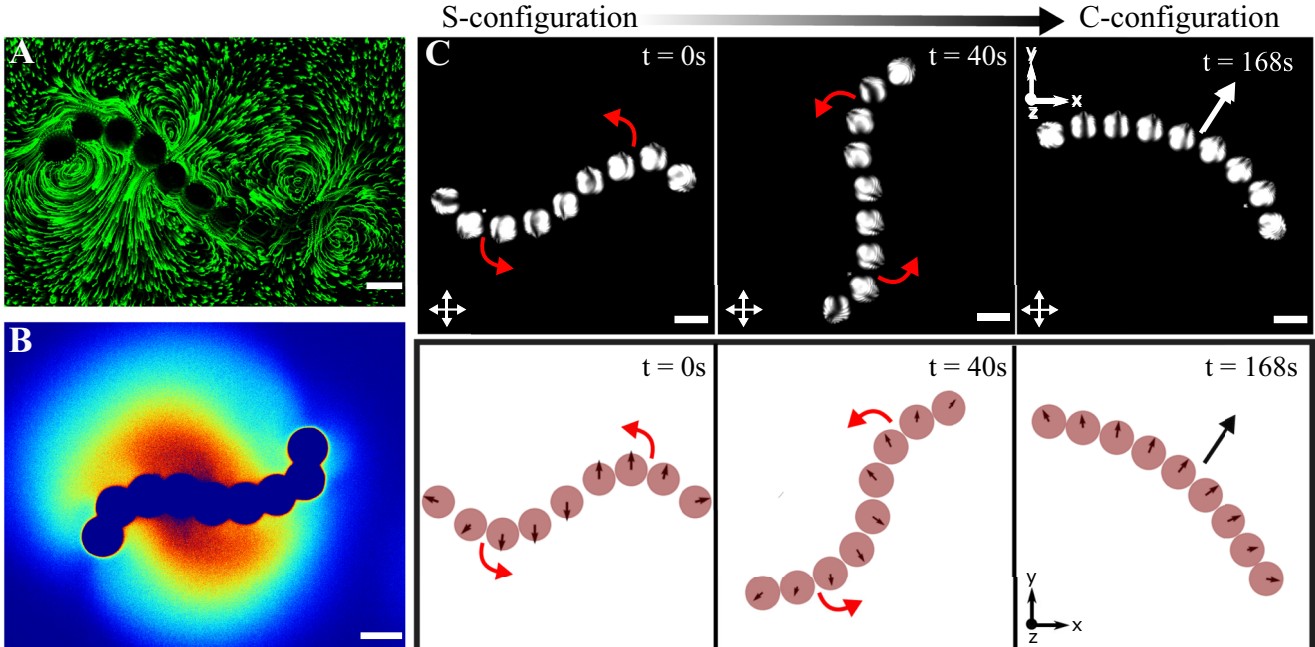

**Fig. 5 | Persistent rotation in a metastable S-shaped configuration and its transition to a stable C-shaped configuration.** Longer active polymers transiently adopt S-shaped configurations with associated characteristic **A** hydrodynamic flow fields and **B** symmetry broken chiral chemical fields (Supplementary Videos SV13 and SV14). **C** In this metastable configuration the polymers undergo persistent rotation and transition spontaneously into the stable C-shaped configuration. Top: The top panel (*left to right*) shows time snapshots of cross-polarised microscopy images of active polymer chain configurations (all scale bars are 50 μm). The instantaneous propulsion directions of droplets in the chain can be visualised with the help of a nematic director pointing in the propulsion direction. The initial time (*t* = 0*s* and 40*s*) frames show that active chain rotation in transient S-configuration (curved red arrows show direction of instantaneous rotation). At *t* ≈ 168*s*, the active chain completely transitions to a stable C-configuration and propels along a selected direction (indicated by the white arrow). Note that the transition time (from S to C-configuration) can be different for different chain lengths. Bottom: The above phenomena are reproduced in our simulations. We see the rotation (curved red arrows) in the S-configuration (*t* = 0*s* and *t* = 40*s*), and a purely linearly translating (long black arrow) rigid chain at *t* = 168*s*.

S-shaped symmetric configuration (Fig. 5 along with Supplementary Videos SV13 and SV14).

In the S-configuration, the active chains exhibit a broken rotational symmetry in the hydrodynamic and chemical fields and undergo spontaneous rotational motion (Fig. 5A, B, Supplementary Videos SV13 and SV14) In these configurations, the chain polarization is discontinuous, with two halves of the chain having polarizations of opposite directions, separated by a defect (discontinuity). As a consequence of the trail dependence on the dynamics, the dynamics at time *t* requires a collective memory of the past trajectories at times *t′* < *t*. This introduces a forward-backward symmetry breaking in the dynamics, where the monomer orientations are guided not only by their instantaneous distances from other monomers along the chain, but also those from previous times *i.e.* history matters in determining the dynamics. The S-configuration is only thus transiently stable, switching back to the C-configuration via a series of monomer reorganizations (Fig. 5C, *top panel*, Supplementary Videos SV15 and SV16). Simulations of our model match this dynamical behaviour (both the polymer rotation and spontaneous transition to the C-shape) and show that, indeed, the dynamics of the S-configuration and its transition are due to interactions with the self-generated chemical field (Fig. 5C, *bottom panel*).

This S-to-C transition is indicative of a broader result, namely that the C-shape is *universal*, attained irrespective of initial conditions chosen. Videos of these for other arbitrary initial conditions are shown in Supplementary Videos SV17 (simulations) and SV18 (experiments). This universality arises due to the collective memory of the past (trail-mediation). Indeed, from our model, we see that this arises for a selected regime of time (length) scales. There are three important time (length) scales that arise from our model. They are a spontaneous propulsion time scale $\tau = \frac{b}{v_s}$, a deterministic response time scale, $\tau_r = \frac{b^3}{\chi_r}$,

and a chemical diffusion time scale $\tau_c = \frac{b^2}{D_c}$ (equivalently $l = b$, $l_r = \frac{v_s b^2}{\chi_r}$ and $l_c = \frac{D_c}{v_s}$). The necessary regime for the universality to hold is $\tau_c < \tau$ and $\tau_r < \tau$ (or equivalently $l_c > b$ and $l_r < b$), such that filled micelles are sufficiently diffuse in the system for the interactions to be in effect (trail can be felt) and that deterministic responses dominate over spontaneous propulsion. In the absence of these, the monomers can be said to "escape" their own trail, and no emergent dynamics will be seen. Indeed, we see that the values that we use in this paper (see Supplementary Information Table 1) are $\tau \approx 0.435s$, $\tau_r \approx 5.8 \times 10^{-5}s$, and $\tau_c \approx 4.0 \times 10^{-4}s$, thus $\tau_r < \tau_c < \tau$, and hence our condition is satisfied.

Given the symmetry breaking induced by the trail-dependence, the C-shape is exclusively selected (i.e. no other stereotypic steady-state shape is seen under confinement) via the following mechanism. The orientation of monomers near the center of the chain will always be aligned perpendicular to the body axis, and hence along the propulsion direction. This is due to their mutual repulsion from monomers either side (see snapshots in Fig. 3). Moving away from the central monomer, orientations are further repelled from this central direction, in proportion to their distance from the central monomer. Thus, the spatial distribution of the chain is such that the central parts of the chain lead the edges, with continuously varying monomer orientations, giving the C-shape.

Our results so far suggest that the emergent dynamics of the flexible active polymers—rigidity, stereoscopic shapes and propulsion—can be qualitatively (and to some extent quantitatively) captured by considering only the effects of confinement and chemical interactions between the droplets. This begs the question of what unique role, if any, does hydrodynamics play in the dynamics *i.e.* can it provide substantial qualitative deviation from the above results (beyond mere quantitative corrections), given the strong coupling between the two (Fig. 2). Such coupling can lead to feedback and time-delay effects

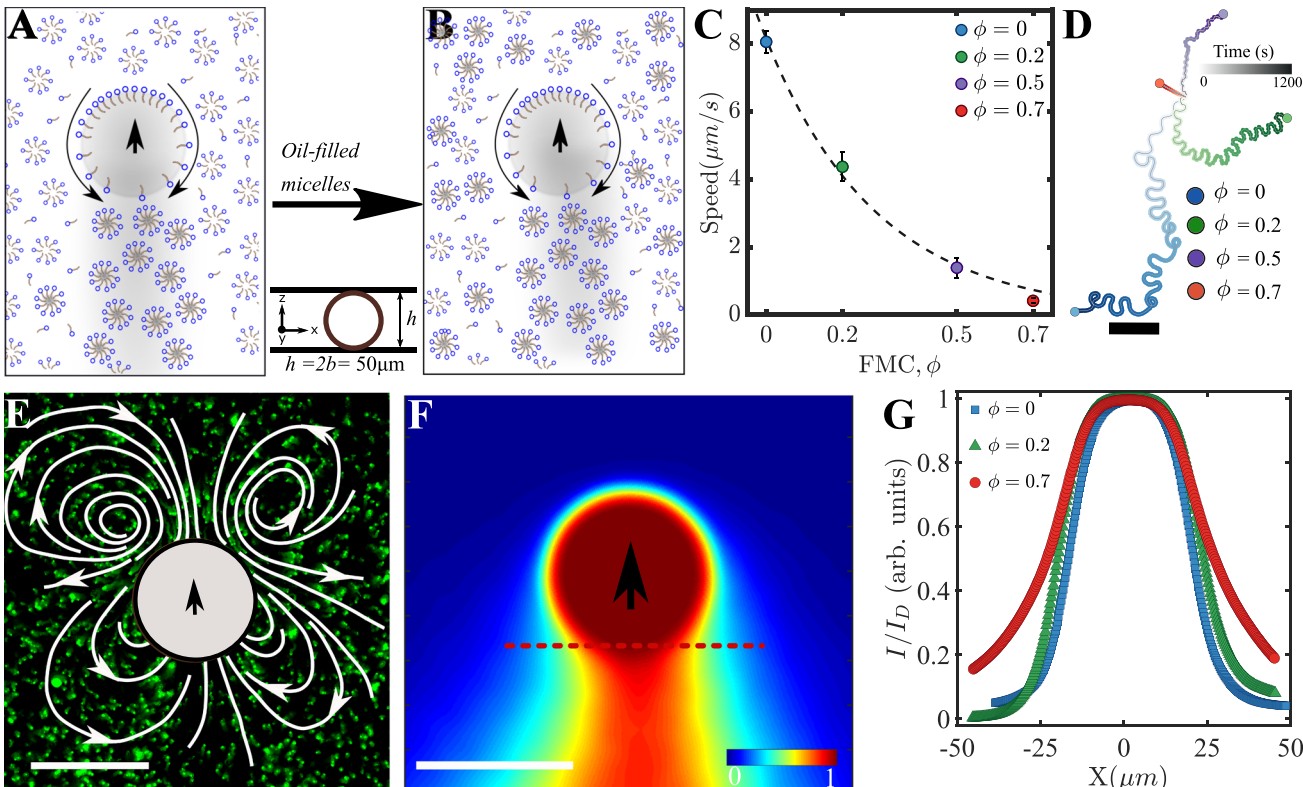

**Fig. 6 | Tuning the chemo-hydrodynamic fields of a monomer.** The schematics showing swimming monomer droplets without additional oil-filled micelles ($\phi = 0$) (**A**) and with additional oil-filled micelles to the swimming droplet environment ($\phi = 0.7$) (**B**). **C** The speed of the monomer droplet is plotted against $\phi$, showing a decrease with increasing concentration of oil-filled micelles (number of droplets, n ≥10 for each concentration of oil-filled micelles, error bars represent ± SD). **D** Experimental trajectories of monomer droplets at a different concentration of oil-filled micelles added to its environment are displayed, color shaded according to time (darker at later times, scale bar: 500 $\mu m$). **E** The contractile nature of the steady-state hydrodynamic field of monomer in a chemically tuned environment with oil-filled micellar solution ($\phi = 0.7$) (scale bar: 50 μm) and **F** the corresponding modified steady-state chemical field (scale bar: 50 μm). **G** The normalised chemical field intensity, $\frac{I}{I_D}$ is plotted along the line normal to the direction of motion of swimming monomer droplets, where $I_D$ is the droplet intensity, $I$ is the intensity of the chemical field. The chemical intensities are reported for $\phi = 0$, $\phi = 0.2$ and $\phi = 0.7$. Source data are provided as a Source Data file.

which in some scenarios can give rise to time-dependent steady-states such as oscillations, beyond what we have hitherto observed. Indeed, hints of such behaviour were already seen in the aforementioned metastability of the dimers. We now experimentally explore such regimes of chemo-hydrodynamic coupling via tuning of the monomer propulsion.

## Tuning the chemo-hydrodynamic fields of the monomer self-propelled droplet

First, we discuss a method to tune the monomer propulsion speed, and then in the next section discuss its implication on the chemo-hydrodynamic coupling. The propulsion of the monomer can be tuned by modulating the transport of fresh surfactant micelles to the droplet interface[42,43,49]. Specifically, such modulation affects the distribution of the surfactant gradient, and thereby the slip velocity on the surface of the droplet, and hence the chemical and hydrodynamic fields. We achieved such tuning by controlling the ratio between fresh ("unfilled") and solubilised-oil laden ("filled") surfactant micelles in the aqueous medium in which the swimmers are embedded (Fig. 6A). The filled micelles are prepared by pre-dissolving 5CB oil droplets in a 25 wt % SDS solution up to saturation. We then adjust the ratio $\phi$, the total fraction of the filled micelles in the solution, using which we obtain a quantitative tuning of the monomer speed (Fig. 6C) and its dynamics (Fig. 6D). It can be seen that with increasing $\phi$, the speed of the monomer reduces sharply. This reduction in speed is associated with qualitative changes in the flow and chemical fields around the monomer (Fig. 6E, F and Supplementary Information Fig. S9A–D). Since our

focus here is to demonstrate substantially novel effects from the coupling between the chemical and hydrodynamic fields, we focus on the case with $\phi = 0.7$ which results in a flow field that is reminiscent of a "puller" type of squirmer (Fig. 6E) compared to the "pusher" type of squirmer at $\phi = 0$ and $\phi = 0.2$. Due to the reduced speed of the droplet in this configuration, the associated chemical field is able to effectively diffuse further, and form a wider trail behind the droplet compared to the less wide trail of monomer observed at $\phi = 0$ and $\phi = 0.2$ (Fig. 6F, G and Supplementary Information Fig. S9C, D). In addition, due to the nature of the flow field, the filled micelles are transported (advected) to the front of the droplet, thus changing time-scales associated with the self-interaction of the monomers with their own chemical trails.

## Time varying chemo-hydrodynamic fields and the emergence of periodic polymer oscillations

We study the dynamics of the chains in this condition *i.e.* $\phi = 0.7$. There are stark differences in the chemical and hydrodynamic flow-fields around the polymers (see Fig. 7, in comparison to Fig. 2). Specifically, clear asymmetries in the fields, *along the length of the chain*, appear in a time-snapshot and these represent time-dependent dynamics (Supplementary Videos SV19–SV20) – these asymmetries are also reflected in the orientations of the monomers within the chains. We further see that these asymmetries can support time-periodic motion (oscillations) in our system. As an illustrative case, we focus on the dynamics of trimers. We report *oscillatory gaits* of the trimers—this can be seen from the trajectories (Fig. 8A, Supplementary Video SV21). By measuring the time-varying chemical fields around the trimer, we

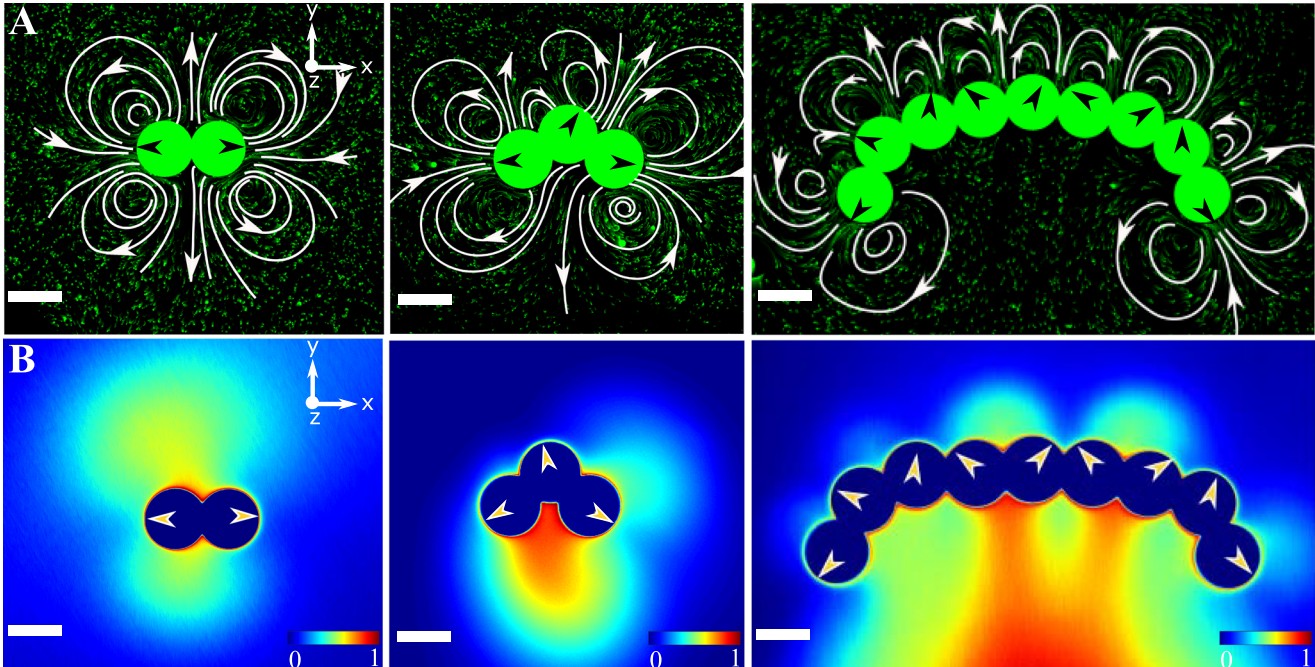

**Fig. 7 | Self-shaping hydrodynamic and chemical fields of active polymer chains in a chemically tuned environment. A** Experimentally measured hydrodynamic flow fields (Supplementary Section PIV, FlowTrace, and SI Fig. S4(C, D, E)). From *left* *to right*: dimer ($N = 2$), trimer ($N = 3$), and nonamer ($N = 9$). **B** Experimentally measured chemical fields (Supplementary Section CF, SI Fig. S4A, B). From *left to right*: dimer, trimer, and nonamer (scale bar: 50 μm).

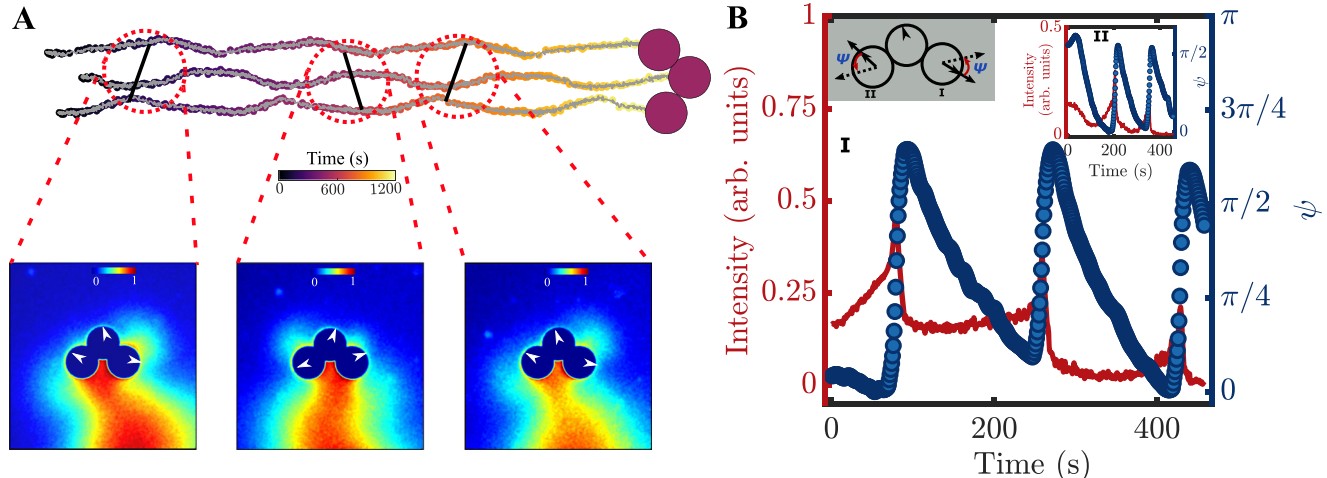

**Fig. 8 | Oscillatory dynamics of active chain in a chemically tuned environment. A** Oscillatory trajectories of a trimer and the time-dependent asymmetry in the corresponding chemical field in a strong two-dimensional confinement and chemically tuned environment. The snapshots of the chemical field show the time-dependent asymmetry in the chemical field. **B** Time-dependent periodic changes are observed in chemical field intensities measured along the droplet orientations. The periodic changes in the droplet orientations (for both edge droplets of the trimer, see inset *top left*) are measured for the droplet at the edge in reference to initial orientations as represented for edge droplet I (*top left*), $\psi$ is the angle measuring the droplet orientation with respect the orientation of the edge droplet farthest from the propulsion direction (schematic, *top left* I). The initial $\psi$ value is different in the inset plot II (*top right*) compared to the plot I, due to different initial orientations ($t = 0$) for both edge droplets from their reference orientation (I). Source data are provided as a Source Data file.

rationalise that these oscillations are caused by the repulsive auto-chemotactic nature of the droplets. It can be seen that the interplay between the flow and chemical fields leads to an asymmetric build-up of the chemical field in front of the chain (Fig. 8A, *bottom panel* and Fig. 7B, *bottom panel*) due to which the monomer self-propulsion direction undergoes a re-orientation. These oscillations are further seen to be periodic—this can be seen clearly from the quantification of the intensity and orientational angle measurements $\psi$, which periodically change on either side of the trimer (Fig. 8B and *insets*). The sharp transitions are suggestive of threshold concentrations of the chemical field that cause fast re-orientations of the monomers. It is also to be noted that the initial $\psi$ values are different for both edge droplets due to their different initial orientations (Fig. 8B, *insets* I and II).

## Discussion
Using chemo-hydrodynamically active monomer droplets, we constructed freely jointed polymers; not only are the joints between the monomers fully flexible but the self-propulsion direction of each

monomer can also freely evolve. The self-interactions within the polymer give rise to emergent self-organization, particularly self-propulsion—stable translation and metastable persistent rotations—of the polymers in rigid configurations. We quantitatively mapped these interactions and using a simple model based on chemical interactions alone, identified minimal, generic conditions in which such novel self-organization can arise—namely trail-mediated chemical self-interactions, and quasi two-dimensional confinement. It must be noted that the active chains in our experiments self-propel in a direction normal to their body axis, which is the direction of maximum drag for a slender body; this is in contrast to natural objects such as microbes and active polymers which typically propel along their major body axis *i.e.* the direction of minimum drag.

The "dry" chemical active matter model we have considered here is in line with similar recent successful descriptions used to describe the scattering dynamics of monomer droplets due to their repulsive auto-chemotaxis[12,50]. However, the effects of hydrodynamic coupling are apparent in our system at high filled micelle concentrations, where we have further found that the monomer self-propulsion speed is suppressed[42,43]. While such coupling does not qualitatively effect the emergent rigidity and self-propulsion that we have reported here, feedback and time-delay effects from such coupling can give rise to steady-states such as oscillations. Studying these dynamics in detail —for example, to uncover the mechanism for oscillations of a trimer— suggest directions for future work.

The emergent dynamics we have reported are in quasi two-dimensional settings. As we have seen, relaxation of this criterion results in a reduction of the rigidity and affects the stability of the stereotypical polymer configurations. For example, when the height of the Hele-Shaw cell $h > 2b$, there is room for extra conformational degrees of freedom for the freely jointed chain. The fully three-dimensional dynamics of active polymers with chemical and hydrodynamic interactions should hold rich possibilities for future exploration. Further, while we have only explored short linear assemblies here, the extension to longer polymers and higher dimensional assemblies such as sheets[21] with more tunable bonds ("multiflavored assemblies"[33]), complex shapes and thereby a richer dynamical repertoire is possible. Such multicomponent monomers with attractive, repulsive and even non-reciprocal interactions[13] may be used to create functional self-morphing assemblies. Finally, the system of emulsion droplets considered here, in contrast to colloidal systems, offers possible modifications in both controlling the internal chemistry of the droplets (to tune their activity) and the flexibility of the assemblies, because of the mobile nature of the droplet surface. We envisage that these systems could also be a step in the direction of designing the multicomponent and multistimuli active delivery systems[51].

## Methods
### Materials
We used 5CB (4-cyno-4'-pentylbiphenyl) liquid crystal procured from Frinton Laboratories, Inc. and SDS (sodium dodecyl sulphate) surfactant procured from Sigma Aldrich. DOPC (1,2-dioleoyl-sn-glcero-3-phosphocoline), Liss-Rhod-PE (1,2-dioleoyl-sn-glycero-phosphoethanolamine-N-(lissamine B sulfonyl) (ammonium salt) and biotinyl-cap-PE (1,2-dioleoyl-sn-glycero-phosphoethanolamine-N-(cap biotinyl) (sodium salt) were purchased from Avanti Polar. Alexa Fluor®488 conjugated streptavidin was procured from Sigma-Aldrich. Fluo-Spheres™ carboxylate-modified Microspheres, yellow-green fluorescent (505/515 nm) and Nile red dye were purchased from Invitrogen (by ThermoFisher Scientific). All chemicals were used as received.

### Preparation of (Lipid (DOPC)—surfactant (SDS)) micellar solution
All lipid stocks were prepared in chloroform: DOPC at 25 mg/mL, Biotinyl-Cap-PE at 10 μg/mL, and Liss-Rhod-PE at 10 μg/mL. We

aliquoted 25 μL of DOPC from the stock, 2 μL of Biot-Cap-PE, and 10 μL of Liss-Rhod-PE in a 2 mL Eppendorf tube to prepare the aqueous phase. The lipids were vacuum dried overnight and then nitrogen dried before being mixed with the aqueous phase. To the tube containing dried lipids, we added 1 mL of 0.125 (w/v)% SDS solution prepared with Milli-Q water. The tube was then left at room temperature ($T = 25$ °C) for lipid hydration. A probe sonicator (VC 750 (750W), stepped microtip diameter: 3 *mm*) with an on/off pulse of 1.5 s/1.5 s and 32% of the maximum sonicator amplitude was used to mix the components. The tube was placed in an ice bath during the sonication steps (sonication for 30 s with a 1 min gap) to avoid overheating of the sample.

### Droplet production
We used an oil-in-water (O/W) emulsion system to make the droplets. 5CB liquid crystal droplets (oil phase) were stabilised using micellar solution (surfactant and lipids) in the aqueous phase. When 5CB LC (the oil phase) is injected through a microfluidic channel and encounters the aqueous phase (an aqueous solution of lipids and surfactant, SI Fig. S1A), adsorption of free lipid-surfactant results in stable 5CB droplets. The monodisperse droplets generated by the microfluidic device were collected and stored in 0.25% SDS solution, where they remained stable for months (SI Fig. S1C, *left panel*). The presence of lipids in the monolayer of lipids and surfactant stabilising the 5CB emulsion droplets was confirmed using fluorescence microscopy (SI Fig. S1C, *middle panel*). Likewise, streptavidin functionalization of biotinylated 5CB droplets was also confirmed using fluorescence microscopy (SI Fig. S1C, *right panel*). Details of the fabrication of microfluidic devices can be found in the Supplementary Information, SI Fig. S1A, and the experimental section (DF).

### Preparation of droplet assemblies
We used a simple method to prepare the 5CB droplet assemblies: we split the sample into two populations. Only biotinylated 5CB droplets were used in population "I" (SI Fig. S1B, a) and streptavidin functionalised biotinylated 5CB droplets in population "II" (SI Fig. S1B, b, c, *right panel*). To remove free biotinylated lipids and streptavidin from the bulk, both populations (I and II) were washed 2-3 times with 0.25% SDS solution. The population "I" is then centrifuged at 1610 × g for 30 s to settle down all the droplets at the bottom of the tube and remove the solvent from the top. On top of the settled fraction of the droplets, we gently added 200 μL of fresh 0.25% SDS solution. The tube was then filled with 5 μL of a 50 mM NaCl salt solution prepared in 0.25% SDS solution, and it was centrifuged for 30 s (at 1610 × g). We then gently added a 20 μL volume of 5CB biotinylated droplets functionalized with streptavidin from population "II" to population "I" at the bottom of the tube (SI Fig. S1B). The tube is then centrifuged at 12,400 × g for 35 min in an Eppendorf centrifuge (Eppendorf centrifuge model 5418) at T = 15 °C. The assemblies were studied inside a quasi two-dimensional flow cell (SI Fig. S1E); the details of the flow cell fabrication are discussed in the experimental section of the Supporting Information (FlowCell).

### Imaging (bright field and fluorescence microscopy)
We used both multi-channel epifluorescence and brightfield microscopy to visualize 5CB emulsion droplets and their linear assemblies. The assemblies were imaged on an Olympus IX81 microscope with a ×4, ×10, and ×20 UPLSAPO objective, and images were captured with a Photometrics Prime camera. For the fluorescence images, we used a CoolLED PE-4000 lamp. A 16-bit multichannel image of 2048 × 2048 pixels was taken, consisting of a green channel (excitation: 460 nm, dichroic: Sem-rock's Quad Band), a red channel (excitation: 550 *nm*, dichroic: Sem-rock's Quad Band), and a bright field channel. We recorded the images and videos using Olympus cellSens software. We used a Leica bright field microscope (model M205FA) equipped with

Leica-DFC9000GT camera to record longer time videos of the linear assemblies. Videos are recorded with a microscope zoom of 2, an exposure time of 0.02 s, and a 1 fps frame rate. We captured a 16-bit image of 2048 × 2048 pixels using a Leica-DFC9000GT-VSC06748 camera.

## Data availability
Experimental data is provided in the source data file. No additional data was used besides the results described in the text. Additional summary statistics and the source code for the plots may be available upon request to the corresponding authors. Source data are provided with this paper.

## Code availability
All code used to produce the figures in the main text are available in the Code Ocean submission along with this paper, from the link https://codeocean.com/capsule/6199254/tree.

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

## Acknowledgements

We acknowledge discussions with Sriram Ramaswamy, Ignacio Pagonabarraga, and Ronojoy Adhikari. We thank Abhrajit Laskar for his help with the numerical computation of the flow fields. We acknowledge support from the Department of Atomic Energy (India), under project no. RTI4006, the Simons Foundation (Grant no. 287975), the Human Frontier Science Program, and the Max Planck Society through a Max-Planck-Partner-Group. The work was funded in part by the Indian Institute of Technology, Madras, India through their seed and initiation grants and Start-up Research Grant (SERB File Number: SRG/2022/000682), SERB, India, to RS.

## Author contributions

S.T., R.S., and M.K. conceived and designed the study. With help from S.T., M.K. and A.M. performed the experiments and data analysis. R.S. and A.G.S. designed the theoretical model and carried out the simulations. M.K., R.S., A.G.S., and S.T. wrote the paper.

## Competing interests

The authors declare no competing interests.
