## [Peer Review File · Nature Communications]

REVIEWER COMMENTS

Reviewer #1 (Remarks to the Author):

The authors create active polymers based on self-propelling oil-in-water-droplets serving as monomers. Based on a well-known mechanism, individual droplets spontaneously break the symmetry in their environment and cause Marangoni flows at the air-water interface, which leads to self-propulsion. The droplets also induce characteristic chemical and hydrodynamic fields leading to interactions of different droplets. As a main novelty compared to previous literature, before activating the droplets by immersing them into a SDS surfactant solution, the authors connect ("polymerize") them in the following way: they use a hybrid monolayer of surfactant and biotinylated lipids that coat the droplets spontaneously and exploit well-known specific interactions of biotin and streptavidin to link the droplets to form chains (of up to about ten monomers without branching). The resulting active polymers are freely-jointed. However, when confined to a quasi 2D setting (Hele-Shaw) the droplet-polymer becomes rather rigid ("C-shape") and propel orthogonal to their long axis. Notably, the authors explicitly measure the flow field (microPIV measurements with fluorescent tracer particles) and the chemical concentration field. For the latter they exploit that the droplets transfer oil into empty micelles, leaving oil-filled micelles behind which diffuse slower than the surrounding empty micelles. The authors visualize the oil-filled surfactant micelles (and hence the relevant concentration field) using an oil-soluble fluorescent dye.

The experimental realization of an active polymer based on linking active droplets is innovative and should be of clear interest both to researchers who are interested in active polymers and in active droplets.

However, there are also significant weaknesses in the manuscript which should be considered in detail, before a recommendation for or against publication in Nature Communications can be made.

In short, the most important drawbacks are that the presentation of the work is not yet convincing and that the experiments and simulations are rather disconnected, with the model not being really convincing at the present stage.

In detail:

1. The article is not yet well written and clearly not at the level expected for Nature Communications.

This concerns in particular the abstract which is confusing and should be rewritten very carefully.

The introduction is also not well written from my perspective; the first sentence is unclear; the remainder does not provide a nice motivation and an exciting literature overview.

Most of the main text is written in an acceptable way, but also there the entire text should be checked and iterated. It contains sentences such as "The orientational rigidity is computer by computing..", which shows that more work on the text is needed.

2. Monomers are represented as point sources for the chemical field in the model. Since the near-field concentration of the polymer determines its self-propulsion, which is the central aspect of the active polymers in question, using point sources and evaluating the field at the mid-points of the droplets seems to be insufficient.

This is also mirrored by the fact, that "for instance, the dipolar field for $N=2$ is not reproduced", which is not good.

3. The model completely neglects hydrodynamics of the droplet-polymers and hence also advection of the micelles in their vicinity.

Is there any evidence that advection of the chemical is weak compared to diffusion? (Given the relatively large size of the micelles and the rather strong flow fields, I would have expected the opposite.)

4. What are the Peclet and the Reynolds number in the present experiments?

5. Simulation parameters: I am missing a clear discussion explaining which parameter values can be deduced from experiments and which ones are chosen more or less arbitrarily chosen. For the latter ones, can

the authors at least give convincing arguments for the chosen orders of magnitude of the parameter values?

6. The results of experiments and simulations are presented almost completely independently of each other. It would

be much stronger to show some quantitative comparison.

7. Why do the droplets not cluster? Why do they form linear chains? What is the physical reason for this?

8. Is there any way to suppress branching and to reach active polymers with $\gg 10$ monomers?

Why does branching occur from this size onwards (only)?

9. Can the authors provide a more detailed explanation of the intuitive mechanism behind the formation of the C-shaped structure, and address why other conformations are not possible in strongly confined systems?

10. Can the authors be more concrete regarding the significance of the active-droplet-polymers for the literature on active polymers?

Can they be used to reproducing some of the generic predictions for active polymers?

11. Can the authors provide some more information on the squirmer simulations of the individual droplets?

I find it somewhat confusing that for these simulations, seemingly only hydrodynamics has been considered, and for the multi-droplet-simulations, hydrodynamics has been fully neglected.

To justify the squirmer model: how large are the flows perpendicular across the droplet surface? I guess that it is a good approximation to neglect them, but this should probably be mentioned and justified.

12. Figure 4 appears rather misleading. While experiments demonstrate a transition from flexible to rigid (C-like shape) structures, simulations show droplets transitioning from rigid-like rods to less rigid states (C-shape) at long times. The authors should reconcile this discrepancy or provide additional clarification to avoid confusion.

13. Figure 2 Caption: The caption for Fig. 2 should probably be revised for clarity. For instance, $h/2b=1$ should probably refer to a strong confinement compared to $h/2b=1.4$, where b represents the droplet radius and h denotes the confinement height(?)

Reviewer #2 (Remarks to the Author):

In their experiments, Kumar et al. investigate the single dynamics of active polymers. They form freely-jointed linear chains of active droplets comprising between $N=2$ and $N=10$ monomers and study their individual behaviour under 2D confinement. They find that the polymers self-propel perpendicular to the chain with a speed that increases with N , and that they take a rigid curved shape. They confront their findings with results from a model that accounts for chemical interactions between droplets, leaving hydrodynamics out.

The results presented in the manuscript are quite intriguing and timely, as the study of active polymers is gaining momentum. In this context, it is a much welcome experimental contribution to the existing body of numerical and theoretical works. For these reasons, I would support publication of this manuscript would they authors satisfactorily address my two major comments below.

Major comments:

1) The presentation of the comparison between experiments and theory should be improved.

a. Speeds (fig 2 H and fig 4 d) should be reported on a single panel. Similarly for curvatures (fig 2 H and fig 2 C and fig 4 E).

b. Comparison of the velocity and chemical fields should be added by plotting cuts along the (xy)-plane.

c. The discussion of the transient of the simulations could be moved to the SI, as it does not bring about important features of the experimental systems. It suffices to say that a steady state is quickly reached in the simulations. This remark concerns the Panels B, C, f, G, H, I of Fig. 4 and the associated paragraph. Skipping this part leaves more room for a quantitative comparison between theory and experiments.

2) On page 9, the authors mention the “puller” nature of the squirmer at $\phi=0.7$. In comparison, in Fig. 1 B, the flow profile is of “pusher” type. Is this correct? Is this change due to the confinement or is there a transition for some finite ϕ value? The authors should state clearly the type of squirmer in Fig 1 B and address any change explicitly when discussing Fig. 6.

Related to this, the dimer flow field in Fig. 3 A resemble the one of two pullers linked together. A comment on this is welcome.

Minor comments:

1) There are many small imprecisions along the text. The authors should proof-read their manuscript before resubmitting. Here are a few examples:

- Bottom of page 2 and Top of page 3: "Furthermore, since these are liquids droplets..." appears two times.

- Caption of Fig. 2: weak confinement is $h/2b = 1.4$ not 1, and vice versa for strong confinement.

- Caption of Fig 2: scale bar is of panel F, not G.

- Caption of Fig 2. D: only the direction perpendicular to the propulsion is plotted.

- Caption of Fig 3: Error bars are mentioned, but no panel have them.

- Page 7: "also" appears twice on line 162 and 163.

- Etc.

2) Could data for all ϕ values reported in Fig. 6 G? And why is the $\phi = 0$ curve symmetric while Fig 1 C is not?

3) Fig. 2 D and E. Which N values do these panels correspond to?

4) ϕ is both the fraction of the filled micelles and solution and an angle in Fig. 8 B. And could this angle be indicated on panel A?

5) Inset in Fig. 8 B is not described.

6) Fig 8. A: Are the chemical fields time-snapshots or averaged of the circles? The representation is not clear and should be improved.

7) Discussion: "polymers which typically propel": (passive) polymers do not typically propel...

NCOMMS-23-63411: Response to reviewers' comments and revision summary

We thank the reviewers for their positive assessment of our manuscript. On the basis of the comments, we have reworked the manuscript, including a substantial rewriting of the manuscript, and modification of the figures. These changes are summarized below, and detailed point-by-point responses are included in the following pages. All the changes that we have made in the main text (editing of existing sentences, addition of new sentences, corrections to typos and major/minor comments) are highlighted in **blue**.

1. We have a completely rewritten abstract and introduction
2. Figures 2, 3 and 4 have swapped places from the previous iteration with major changes, in accordance with the reviewers' suggestions of presenting the experimental and simulations together
3. In accordance with the above changes, we have also restructured the main text to reflect better our modeling choices and the correspondence between simulations and experiments.
4. We have also updated the SI and supplementary videos.

REVIEWER 1 (REMARKS TO THE AUTHOR):

The authors create active polymers based on self-propelling oil-in-water-droplets serving as monomers. Based on a well-known mechanism, individual droplets spontaneously break the symmetry in their environment and cause Marangoni flows at the air-water interface, which leads to self-propulsion. The droplets also induce characteristic chemical and hydrodynamic fields leading to interactions of different droplets. As a main novelty compared to previous literature, before activating the droplets by immersing them into a SDS surfactant solution, the authors connect ("polymerize") them in the following way: they use a hybrid monolayer of surfactant and biotinylated lipids that coat the droplets spontaneously and exploit well-known specific interactions of biotin and streptavidin to link the droplets to form chains (of up to about ten monomers without branching). The resulting active polymers are freely-jointed. However, when confined to a quasi 2D setting (Hele-Shaw) the droplet-polymer becomes rather rigid ("C-shape") and propel orthogonal to their long axis. Notably, the authors explicitly measure the flow field (microPIV measurements with fluorescent tracer particles) and the chemical concentration field. For the latter they exploit that the droplets transfer oil into empty micelles, leaving oil-filled micelles behind which diffuse slower than the surrounding empty micelles. The authors visualize the oil-filled surfactant micelles (and hence the relevant concentration field) using an oil-soluble fluorescent dye.

The experimental realization of an active polymer based on linking active droplets is innovative and should be of clear interest both to researchers who are interested in active polymers and in active droplets. However, there are also significant weaknesses in the manuscript which should be considered in detail, before a recommendation for or against publication in Nature Communications can be made. In short, the most important drawbacks are that the presentation of the work is not yet convincing and that the experiments and simulations are rather disconnected, with the model not being really convincing at the present stage.

We thank the reviewer for noting the novelty and interest of our work. We are also grateful for their critical feedback regarding the presentation of our results, particularly the content of portions of the text and the correspondence between experiments and simulations, which we have worked hard to address in the revised manuscript. Further, we have now detailed, both in the manuscript and the responses below, the choices we have made regarding the model.

1. *The article is not yet well written and clearly not at the level expected for Nature Communications. This concerns in particular the abstract which is confusing and should be rewritten very carefully. The introduction is also not well written from my perspective; the first sentence is unclear; the remainder does not provide a nice motivation and an exciting literature overview. Most of the main text is written in an acceptable way, but also there the entire text should be checked and iterated. It contains sentences such as "The orientational rigidity is computer by computing..", which shows that more work on the text is needed.*

Response: We are very grateful to the reviewer for this comment. Based on the suggestions, we have re-written many sections of the manuscript, including a full rewrite of the abstract and the introduction which we hope is more appealing. In addition, we have carefully checked and corrected the entire manuscript for inaccuracies and poor formulation.

2. *Monomers are represented as point sources for the chemical field in the model. Since the near-field concentration of the polymer determines its self-propulsion, which is the central aspect of the active polymers in question, using point sources and evaluating the field at the mid-points*

of the droplets seems to be insufficient. This is also mirrored by the fact, that “for instance, the dipolar field for $N=2$ is not reproduced”, which is not good.

Response: The reviewer correctly points out that in our modelling we do not include higher order contributions to the chemical field. A full model for our system will include a solution of the chemical and hydrodynamic fields along with satisfying appropriate boundary conditions on the surface of the particles and at all other fluid-solid boundaries. Such a solution will include all modes of the chemical and hydrodynamic fields and would necessitate a full numerical solution. This is a rather challenging task that is beyond the scope of this current work and we postpone it for the future. Instead, our aim here has been to seek a *minimal* model for the experimental observations. We start with a simple model where each particle is modelled as a point emitter of chemical field. It is indeed remarkable that this modeling choice is amply justified *post-facto*: the model correctly predicts the universal and stable C-shape chain seen in the experiments, underscoring the importance of the chemical interaction in these settings. In addition, our predictions for the MSD, curvature, and speed of the chain using this simple model are found to be in good agreement with experimental data (see new Fig 3 and Fig 4). While some of our estimates, such as the speed of the C-shaped chain in simulation and the curvature, do not have an *exact* quantitative match with those of experiment, the *qualitative* behaviour - i.e the variation of these quantities with respect an increase in chain length - is well reproduced. This is shown in juxtaposed panels of the new Fig 4. For these reasons, we believe that our minimal model captures the essential features of the experiment, and we will seek an exact quantitative match with the full coupled chemo-hydrodynamics in future work. We have now added an appropriately condensed version of the above comment in the paper. Please see Section C and D, pages 4–6 and new Figs. 3 and 4.

3. *The model completely neglects hydrodynamics of the droplet-polymers and hence also advection of the micelles in their vicinity. Is there any evidence that advection of the chemical is weak compared to diffusion? (Given the relatively large size of the micelles and the rather strong flow fields, I would have expected the opposite.)*

Response: The reviewer has again pointed very correctly on our model which ignores hydrodynamic interactions. As mentioned in the response to the previous point, the model used in the manuscript should be thought of as a minimal description with the aim of capturing essential features of the experimentally observed dynamics. Indeed, the full problem involves simultaneous solutions of the chemical and hydrodynamic field along with appropriate boundary conditions. This is a challenging numerical problem which we postpone to a future work. At the same time, here we provide an argument to justify our choice to ignore the hydrodynamic interactions *i.e.* that in strong confinements within a Hele-Shaw cell, chemical fields decay more slowly than hydrodynamic fields. In strong confinements, when the height of the Hele-Shaw cell is of the order of the diameter of the particle, the chemical field is known to decay as $1/r$, while the leading advective fluid flows due to activity are known to decay $1/r^3$ (see, for example, E Kanso and S Michelin, *J. Chem. Phys.* 150, 044902 (2019)). Advection by the fluid flow may be therefore ignored to the leading order. Thus, in keeping with the minimal nature of the model, we consider only chemical interactions for our model. Please see the text in Section C.

4. *What are the Peclet and the Reynolds number in the present experiments?*

Response: We have computed the Peclet number $Pe = \frac{vb}{D_c}$, where the diffusivity of empty micelles is, $D = 28.03 \mu\text{m}^2/\text{s}$ (data provided in Fig. 1 below). The Peclet number is $Pe \approx 7$ and Reynolds number is $Re \approx 0.7 \times 10^{-4}$ where density is $\rho = 1.0043 \text{ g/ml}$, radius of the droplet $b =$

FIG. 1. Diffusion of empty micelles (*red color*) and diffusion of oil-filled micelles (*blue color*) at different concentrations of filled micelles (FMC), ϕ .

25 μm , speed $v = 8 \mu\text{m}/\text{s}$, and viscosity $\eta = 5.2 \text{ m Pa}\cdot\text{s}$. All the measurements were made at room temperature ($T = 25^\circ \text{ C}$).

5. *Simulation parameters: I am missing a clear discussion explaining which parameter values can be deduced from experiments and which ones are chosen more or less arbitrarily chosen. For the latter ones, can the authors at least give convincing arguments for the chosen orders of magnitude of the parameter values?*

Response: We thank the reviewer for this comment and agree that this information was previously unclear (they were previously only mentioned in the SI). The colloidal length and propulsion speed are identical to the experiment. The diffusivity that we have used is 50 times higher than in the experiment. An identical value for the diffusivity would also give the same phenomenology, though the curvature of the C-shape is much less pronounced. Thus, the pronounced curvature of the C-shape can be attributed to the micellar advection, which we account for via this parameter choice. A full study of the effect of different parameter choices (including diffusivity) on the dynamics of the chain is presented in a follow-up work (AG Subramaniam, M Kumar, S Thutupalli, R Singh, *arXiv:2401.14178* (2024)). We have also now stated in the main text that the only two parameters that are arbitrarily chosen are the susceptibilities (rotational) χ_r and (translational) χ_t - these determine how much monomers would preferentially rotate and translate away from each other in response to chemical gradients - indeed currently these effects are also difficult to measure from the experiments directly. We have also now included in the text a discussion of the relevant time and length scales in the problem and specified in which regime the emergent rigidity

occurs. The full list of parameters used are available in SI Table 1. Please see the text in Section C.

6. *The results of experiments and simulations are presented almost completely independently of each other. It would be much stronger to show some quantitative comparison.*

Response: We thank the reviewer for this suggestion. We have now reorganized the figures to present the experiments and simulations together. Therefore, we now have modified Figures 3 and 4 to show experimental data together with simulations. This is accompanied by a major rewriting of the text which brings out the salient aspects of the model and the comparison of the simulations with the experiments.

7. *Why do the droplets not cluster? Why do they form linear chains? What is the physical reason for this?*

Response: We address this at two levels: (i) the level of construction of the assemblies, which is an equilibrium process, and (ii) when the assemblies are rendered active *i.e.* driven out of equilibrium. At the level of construction of the assemblies, this is addressed in the response of the next comment (8). Once the assemblies are formed and are further driven out-of-equilibrium, droplets in a given chain do not cluster due to their (auto)-chemorepulsive nature which naturally precludes droplets from coming close to each other (as is required for clustering). Clustering would occur if the interactions were chemo-attractive ($\chi_r < 0$ and $\chi_t < 0$). We have indeed explored such situations in a recent follow up work (AG Subramaniam, M Kumar, S Thutupalli, R Singh, *arXiv:2401.14178* (2024)).

8. *Is there any way to suppress branching and to reach active polymers with > 10 monomers? Why does branching occur from this size onwards (only)?*

Response: This indeed would be possible by using linear microfluidic channels as a trap where droplets can be chemically linked to create controlled-length and longer chain-length linear active polymers. Another possibility would be to create an acoustic trap for the droplets. We are already exploring these techniques to create controlled length structures. Given that the assembly process is a one-pot reaction in equilibrium, the monomers link with each other only probabilistically in time; intuitively, therefore, the chance of branching of linear chains increases with increasing chain length. Notably, we also observed branching even for shorter assemblies (monomers $N \leq 10$), but the yields of linear chains are higher and they are also easily separable from the branched chains and free monomers in our protocol. The branching of the linear chains or clustering would also be controlled by the fraction of the streptavidin and biotinylated lipids in the multicomponent monolayer stabilising the droplets. However, a detailed investigation is still required to explore the effects of chemical components linking the droplets, which we hope to take up in the future.

9. *Can the authors provide a more detailed explanation of the intuitive mechanism behind the formation of the C-shaped structure, and address why other conformations are not possible in strongly confined systems?*

Response: We thank the reviewer for this comment, and now we have included a brief discussion of this mechanism in the text. We have also included a comparison of time (and length) scales of the problem, and specified the regime where the C-shape is obtained. In addition, we have emphasized the role of *trail mediation* - the collective memory of the past trajectories - that induce a forward-backward symmetry breaking in the dynamics. We have in addition shown that

this topology is *universal* due to this trail mediation. This is discussed in Section E and SI Video 17, 18. Other confirmations (*e.g.* the S shape discussed in Section E) exist transiently - which arise for example due to collision with other chains - and transition to the ballistically propelling C-shape in a stable steady-state.

10. *Can the authors be more concrete regarding the significance of the active-droplet-polymers for the literature on active polymers? Can they be used to reproducing some of the generic predictions for active polymers?*

Response: We thank the reviewer very much for these questions. The field of active polymers has indeed been quite vibrant and the inspiration for this field predominantly comes from biological active polymers such as actin and microtubules and their emergent mechanics. There has also been some reference in the literature to “synthetic active polymers”. However, as we have now clearly stated in the rewritten introduction, except for one realisation (Biswas *et. al.* ACS Nano (2017)), all other experimental realisations of active polymers has been to externally activate isolated colloids and it is the external actuation that assembles the colloids into structures which resemble chains. A few examples of such works are provided below:

- 1) Judit Clopés Llahí, Aitor Martín-Gómez, Gerhard Gompper, and Roland G. Winkler, Simulating wet active polymers by multiparticle collision dynamics, Phys. Rev. E 105, 015310 (2022)
- 2) F.Martinez-Pedrero, A.Ortiz-Ambriz, I.Pagonabarraga, and P.Tierno, “Colloidal microworms propelling via a cooperative hydrodynamic conveyor belt,” Phys. Rev. Lett. 115, 138301 (2015)
- 3) D.Nishiguchi, J.Iwasawa, H.-R.Jiang, and M.Sano, “Flagellar dynamics of chains of active Janus particles fueled by an AC electric field,” New J. Phys.20, 015002 (2018).
- 4) J.Zhang, J.Yan, and S.Granick, “Directed self-assembly pathways of active colloidal clusters,” Angew. Chem., Int. Ed.55, 5166 (2016)

In the absence of the external drive, the structure in the above examples is no longer stable and has no dynamics. This is in stark contrast to a real polymer in which monomers are linked to each other via bonds. Our system is first to achieve a chemically linked polymer in which each monomer unit is independently active.

However, there have been many theoretical (simulation) studies that consider active polymer models similar to the ones that we have constructed. For example: A.Martín-Gómez *et. al.*, “Active Brownian filaments with hydrodynamic interactions: Conformations and dynamics,” Soft Matter 15, 3957 (2019). Our experimental system therefore represents the first experimental realization of such active polymer models. Despite these, we do see important connections to existing generic predictions for active polymers such as self-propulsion. Further, in the present study, we have restricted ourselves to short chains and studying their dynamics in 2 dimensions. Relaxing both these criteria will allow us to uncover many more scenarios which have been considered in literature so far – however it must be noted that the coupling between chemical and hydrodynamic fields as present in our system is novel and may therefore lead to even richer dynamics than predicted.

Finally, we note some of the significant aspects of the active-droplet polymers that we have created:

- (1) Active-droplet-polymers offer a well-controlled, simplified, microscopic system. It provides

great experimental ease and control for measuring the dynamics.

- (2) Each monomer unit in a chain is active and has autonomous motion in a fluid medium.
- (3) The soft droplet interfaces allow easy chemical modifications and thereby freely rearrange even after binding.
- (4) This simplified active-droplet-polymers system can lead to complex emergent dynamics due to the chemo-hydrodynamic interactions. Since the droplets are linked flexibly, it is possible to generate foldable active structures with the assemblies.
- (5) The dynamics can be tuned due to external changes, for example, chemical and physical that we have already demonstrated to some extent in our present study.
- (6) The assembly protocol that we have developed is generic and therefore allows us to create not only chains but also other branched, closed structures and even two and three dimensional assemblies such as sheets and solids.

We have now summarised many of these aspects in both the introduction and the discussion sections of the manuscript.

11. Can the authors provide some more information on the squirmer simulations of the individual droplets? I find it somewhat confusing that for these simulations, seemingly only hydrodynamics has been considered, and for the multi-droplet-simulations, hydrodynamics has been fully neglected. To justify the squirmer model: how large are the flows perpendicular across the droplet surface? I guess that it is a good approximation to neglect them, but this should probably be mentioned and justified.

Response: We thank the reviewer for pointing this out. Indeed, the squirmer simulations are superfluous. We do not include hydrodynamic interactions in our minimal model of the experimental system. A reasoning for this is already present in response to comment 3 of the referee. Thus, we have removed the numerically computed flow plots using squirmer model from Figures 1B and 6E of the older version of the manuscript to preempt any confusion and to keep our discussions consistent.

12. Figure 4 appears rather misleading. While experiments demonstrate a transition from flexible to rigid (C-like shape) structures, simulations show droplets transitioning from rigid-like rods to less rigid states (C-shape) at long times. The authors should reconcile this discrepancy or provide additional clarification to avoid confusion.

Response: We thank the reviewer for this observation, and indeed in the current comparison of the snapshots in Fig. 3 we have almost identical initial conditions to that of the experiment. We note that the transition from a flexible chain to the C-like shape is indicative of the effect of confinement (Fig. 2A and B), not initial conditions. We now further show that the C-shape is universal, and is selected for any initial condition (see Response 9 and Supplementary Video 17). When we observe these active chains in experiments, even if the chain is in any random initial configuration, it immediately transitions to a steady state C-like shape by rearrangement of monomers. This can be seen in multiple instances, for example, S-C transitions shown in Fig 5 (Supplementary Videos SV12 and SV15) where active chain colloids with surrounding assemblies undergo transitions to a series of random configurations and maintain the stable C-like shape.

FIG. 2. Experimental snapshots of an active chain (under a cross-polarised microscope) showing the transition from an initially transient rod-like shape to a stable C-configuration in steady state (Supplementary Video SV18).

Furthermore, the time snapshots of the active chain with an initial rod-like shape transitions to a stable C-like shape (See Fig S8 of SI, Supplementary Videos SV17 and SV18). This shows that even if the chain is in any initial configuration under strong confinement, it always attains the C-like steady state both in experiments and simulations. As mentioned in response 9 (see also Section E of main text), this universality holds for a specific regime of time (length) scales of the system. For the corresponding changes discussing this universality, please see the text in Section E

13. *Figure 2 Caption: The caption for Fig. 2 should probably be revised for clarity. For instance, $h/2b=1$ should probably refer to strong confinement compared to $h/2b=1.4$, where b represents the droplet radius and h denotes the confinement height(?)*

Response: We thank the reviewer for pointing out this oversight. We have now made the appropriate changes in the Figs. captions.

REVIEWER 2 (REMARKS TO THE AUTHOR):

In their experiments, Kumar et al. investigate the single dynamics of active polymers. They form freely-jointed linear chains of active droplets comprising between $N=2$ and $N=10$ monomers and study their individual behaviour under 2D confinement. They find that the polymers self-propel perpendicular to the chain with a speed that increases with N , and that they take a rigid curved shape. They confront their findings with results from a model that accounts for chemical interactions between droplets, leaving hydrodynamics out.

The results presented in the manuscript are quite intriguing and timely, as the study of active polymers is gaining momentum. In this context, it is a much welcome experimental contribution to the existing body of numerical and theoretical works. For these reasons, I would support publication of this manuscript would they authors satisfactorily address my two major comments below.

We thank the reviewer for their very positive assessment of our work and supporting its publication. We are also grateful for their many useful comments and suggested improvements which we have tried our best to address.

Major comments:

- 1) The presentation of the comparison between experiments and theory should be improved.
 - a. Speeds (fig 2 H and fig 4 d) should be reported on a single panel. Similarly for curvatures (fig 2 H and fig 2 C and fig 4 E).
 - b. Comparison of the velocity and chemical fields should be added by plotting cuts along the (xy)-plane.
 - c. The discussion of the transient of the simulations could be moved to the SI, as it does not bring about important features of the experimental systems. It suffices to say that a steady state is quickly reached in the simulations. This remark concerns the Panels B, C, f, G, H, I of Fig. 4 and the associated paragraph. Skipping this part leaves more room for a quantitative comparison between theory and experiments.

Response: We thank the reviewer for these comments.

- (a) We agree and have re-organized our figures. Comparison of curvature, orientational and positional rigidities are reported in Figure 3, while MSD and saturation speed are now reported on a single figure (Figure 4), with experiment and simulation on top and bottom panels respectively.
- (b) For the experimental fields, we note that such a cut has been already shown in Fig. 6G. We note that there are no flow fields in the simulation (we have used a ‘dry’ model). Simulated chemical fields are in SI Fig. S8.
- (c) We agree with this, and we have moved the majority of the snapshots to the SI Fig. S8. We have now shown a full evolution in *bottom panel* of Fig. 3, to be compared with the experimental results in *top panel* of Fig. 3.

2) *On page 9, the authors mention the “puller” nature of the squirmer at $\phi = 0.7$. In comparison, in Fig. 1 B, the flow profile is of “pusher” type. Is this correct? Is this change due to the confinement or is there a transition for some finite ϕ value? The authors should state clearly the type of squirmer in Fig 1 B and address any change explicitly when discussing Fig. 6. Related to this, the dimer flow field in Fig. 3 A resemble the one of two pullers linked together. A comment on this is welcome.*

Response: We thank the reviewer for this comment. Yes, it is correct that the monomer

droplet is the “puller” type at $\phi = 0.7$ (Fig. 6E) and the “pusher” type at $\phi = 0$ (Fig. 1B). This transition from “pusher” type to “puller” is not due to confinements as we used the fixed confinement ($h = 50 \mu m$), but for a finite ϕ value ($\phi = 0.7$). Similar transitions have also been reported in the literature (J. Fluid Mech. 2023, 966, A29.; Soft Matter, 2023, 19, 3783-3793). The dimer is in a metastable symmetric configuration with the two connected droplets sitting in each other’s chemical field (please refer to the corresponding chemical field panel). Due to this configuration with the self-propulsion directions pointed away from each other, the resultant chemicals are emitted from the point where the two droplets are connected and grow along the sides of the droplets just like a monomer chemical profile at $\phi = 0.7$ where it is acting like a puller. But, once dimer breaks this metastable state symmetry, it propels like a pusher. An analogy can be drawn from a monomer to the dimer where droplets are acting like two pullers connected.

Minor comments:

1) *There are many small imprecisions along the text. The authors should proof-read their manuscript before resubmitting. Here are a few examples:*

Response: We thank the reviewer for the critical evaluation of our manuscript and for pointing out problems with the text. We have now had a careful revision which hopefully has taken care of all of these issues.

- *Bottom of page 2 and Top of page 3: “Furthermore, since these are liquids droplets...” appears two times.*

Response: We agree with the reviewer and removed it from the later sentence.

- *Caption of Fig. 2: weak confinement is $h/2b = 1.4$ not 1, and vice versa for strong confinement.*

Response: We have corrected it in our revised manuscript

- *Caption of Fig 2: scale bar is of panel F, not G.*

Response: We have changed it in the Fig.3 caption which is a new corresponding Figure.

- *Caption of Fig 2. D: only the direction perpendicular to the propulsion is plotted.*

Response: We modified the text in the caption of Fig. 3 (new figure) as suggested by the reviewer to – “The positional rigidity measured from the chain centre of the mass only in a direction perpendicular to the direction of propulsion.”

- *Caption of Fig 3: Error bars are mentioned, but no panel have .*

Response: We agree with the reviewer, it is now removed from the Fig.3 caption (old figure) and it is now Fig.2 in the revised manuscript.

- *Page 7: “also” appears twice on line 162 and 163.*

Response: We removed extra “also” from this sentence.

2) Could data for all ϕ values reported in Fig. 6 G? And why is the $\phi = 0$ curve symmetric while Fig 1 C is not?

Response: We report the data for $\phi = 0.2$ in addition to $\phi = 0$ and $\phi = 0.7$ in the modified plot of Fig. 6G and SI Fig. S9. This confusion is caused due to the wrong representations of the cut along which the chemical field intensities are measured. The chemical field intensities are measured along a tangent line normal to the propulsion direction of the droplet. That is how we observed a symmetric curve for both the $\phi = 0$ and $\phi = 0.7$ values. These measurements show the spread of chemical field intensity comparatively more along the surface of the monomer for $\phi = 0.7$ than in $\phi = 0$ case. Please see in the text of Section G and in the caption for Figure 8.

3) Fig. 2 D and E. Which N values do these panels correspond to?

Response: These panels correspond to $N = 7$ monomers, we have also updated it in the Figure caption (Fig 3).

4) ϕ is both the fraction of the filled micelles and solution and an angle in Fig. 8 B. And could this angle be indicated on panel A?

Response: We are representing the ϕ for fraction of filled micelles and ψ for an angle. These corrections are made both for Fig 6 and Fig 8. A schematic is also added in the inset of Fig 8 panel to indicate the angle that is measured for the droplets at either end of a trimer showing oscillatory gaits. Please see the corrections in Figure 6 and 8.

5) Inset in Fig. 8 B is not described.

Response: The inset for Fig 8B is now described both in the caption and main text. It is also to be noted that the initial ψ values are different for both edge droplets due to their different initial orientations (inset, Fig.8A and inset, Fig.8B).

6) Fig 8. A: Are the chemical fields time-snapshots or averaged of the circles? The representation is not clear and should be improved.

Response: The chemical fields are time snapshots not averaged of these circles. These circles were merely used to highlight the asymmetry in the chemical field, which we have now removed from the figure 8A to avoid confusion.

7) Discussion: “polymers which typically propel”: (passive) polymers do not typically propel...

Response: Yes, we agree with the reviewer and it should be an active polymer. In the revised draft, we have changed the text with “active polymer”.

REVIEWER COMMENTS

Reviewer #1 (Remarks to the Author):

The authors have significantly improved the presentation of their manuscript. However, the theoretical model has not been improved in response to my main criticism in the previous report. In addition, I think that the motivation and justification of the minimal model by the following text, is not adequate:

"In building a minimal model, we begin by noting that, specifically for the kind of chemo-hydrodynamic swimmers as we have above, chemical fields decay slower than hydrodynamic fields in two dimensions [45]. Indeed, such a rationalization has been applied elsewhere in successfully describing the dynamics of individual droplets [12]. Therefore in our model for the active polymers, we ignore hydrodynamic effects and account only for the chemical interactions."

As argued before:

Near-field hydrodynamics and in particular near-field phoretic fields (not addressed in the quoted text) are key to understand self-propulsion of droplets and droplet-chains (where individual droplets are almost at contact). While I appreciate that corresponding simulations might be beyond the scope of a simple model, their importance cannot be argued away based on pure by far-field arguments. In addition, in the presented experiments, solute advection is probably much more important than solute diffusion. Despite this, the model exclusively captures solute diffusion and uses a diffusion coefficient which is 50 times too high. The excuse that the used model is a "minimal model" does also not help here. A minimal model should capture the key effects that are needed to describe experiments.

It would be important to at least clarify the situation for the reader, i.e., explain which aspects of the experiments the model describes and which ones not. This would be better than providing a highly incomplete argument which tends to misguide the reader.

Overall, I still believe that the experiments present an original new realization of active polymers and that the overall presentation of the manuscript has improved, while the weaknesses in the modeling and simulation part essentially still prevail.

I leave it to the editors to decide if the manuscript

can be accepted for publication (after improving the motivation and justification of the minimal model) or not.

Reviewer #2 (Remarks to the Author):

The authors have satisfactorily answered my comments and questions. Their revised manuscript presents experimental and theoretical results in parallel, making direct comparisons much easier. I recommend publication of the present manuscript.

Minor comments:

Figure 3: Simulation panels. The snapshots do not seem to match with the Supp. Video: the initial state $t=0$ looks neither straight (as in the Video) or random (as stated in the main text).

Figure 8: the red dashed circles on top of the trajectory would be better replaced by a line indicating the specific instants of the snapshots.

NCOMMS-23-63411: Response to reviewers' comments and revision summary

REVIEWER 1 (REMARKS TO THE AUTHOR):

The authors have significantly improved the presentation of their manuscript. However, the theoretical model has not been improved in response to my main criticism in the previous report. In addition, I think that the motivation and justification of the minimal model by the following text, is not adequate:

"In building a minimal model, we begin by noting that, specifically for the kind of chemo-hydrodynamic swimmers as we have above, chemical fields decay slower than hydrodynamic fields in two dimensions [45]. Indeed, such a rationalization has been applied elsewhere in successfully describing the dynamics of individual droplets [12]. Therefore in our model for the active polymers, we ignore hydrodynamic effects and account only for the chemical interactions." As argued before: Near-field hydrodynamics and in particular near-field phoretic fields (not addressed in the quoted text) are key to understand self-propulsion of droplets and droplet-chains (where individual droplets are almost at contact). While I appreciate that corresponding simulations might be beyond the scope of a simple model, their importance cannot be argued away based on pure by far-field arguments. In addition, in the presented experiments, solute advection is probably much more important than solute diffusion. Despite this, the model exclusively captures solute diffusion and uses a diffusion coefficient which is 50 times too high. The excuse that the used model is a "minimal model" does also not help here. A minimal model should capture the key effects that are needed to describe experiments. It would be important to at least clarify the situation for the reader, i.e., explain which aspects of the experiments the model describes and which ones not. This would be better than providing a highly incomplete argument which tends to misguide the reader.

Overall, I still believe that the experiments present an original new realization of active polymers and that the overall presentation of the manuscript has improved, while the weaknesses in the modeling and simulation part essentially still prevail. I leave it to the editors to decide if the manuscript can be accepted for publication (after improving the motivation and justification of the minimal model) or not.

Response: We are grateful to the reviewer for the comments. Indeed, the referee correctly points out that arguments based on far-field descriptions are not valid for nearby droplets. We have now explicitly emphasized this point in the revised manuscript. We have now re-written Section C accordingly, noting that a full numerical solution would need both near and far-field approaches, which we do not pursue in this paper. We have also explicitly noted that while micellar advection is indeed important, we only include diffusion effects here.

We also particularly thank the referee for appreciating the scope of the model for the current work and we sincerely hope that, regarding the modeling approach, the current writing of the manuscript gives the following clear messages to the reader:

- 1) A model with purely (trail-mediated) chemical interactions reproduce the experimental observations of (i) emergent rigidity under confinement, and (ii) increased stiffening and speeding up of longer chains.
- 2) Phenomena such as chemo-hydrodynamic oscillations are indeed important aspects that are not captured by the modeling framework yet and are promising directions for future work in this space.

REVIEWER 2 (REMARKS TO THE AUTHOR):

The authors have satisfactorily answered my comments and questions. Their revised manuscript presents experimental and theoretical results in parallel, making direct comparisons much easier. I recommend publication of the present manuscript.

We thank the reviewer for their positive assessment of our work and for recommending the current manuscript for publication.

Minor comments:

Figure 3: Simulation panels. The snapshots do not seem to match with the Supp. Video: the initial state $t=0$ looks neither straight (as in the Video) or random (as stated in the main text).

Response: We thank reviewer for pointing out this oversight. The snapshot for $t = 0$ is now updated which shows the initial state in Fig. 3 identical to that of SV10. The larger point that we wish to note is that this steady-state is obtained irrespective of initial conditions, as discussed further in Section E.

Figure 8: the red dashed circles on top of the trajectory would be better replaced by a line indicating the specific instants of the snapshots.

Response: We now added the lines on the trajectory to represent the specific instants of the snapshots in Fig. 8.

REVIEWERS' COMMENTS

Reviewer #1 (Remarks to the Author):

The motivation of the used minimal model has been improved which should help readers to understand the ambition of the model and to put the simulation results into perspective. Overall, I am happy to support publication of this article.